# Splitting Credit Risk into Systemic, Sectorial and Idiosyncratic Components

**Alfonso Novales** [1,2,*] and **Alvaro Chamizo** [3,*]

1   Instituto Complutense de Análisis Económico (ICAE), Facultad de Ciencias Económicas y Empresariales, Campus de Somosaguas, Universidad Complutense, 28223 Madrid, Spain

2   Departamento de Análisis Económico, Facultad de Ciencias Económicas y Empresariales, Campus de Somosaguas, Universidad Complutense, 28223 Madrid, Spain

3   BBVA, Ciudad BBVA—Calle Azul, 4 28050 Madrid, Spain

*   Correspondence: anovales@ccee.ucm.es (A.N.); alvaro.chamizo@bbva.com (A.C.)

**Abstract:** We provide a methodology to estimate a global credit risk factor from credit default swap (CDS) spreads that can be very useful for risk management. The global risk factor (GRF) reproduces quite well the different episodes that have affected the credit market over the sample period. It is highly correlated with standard credit indices, but it contains much higher explanatory power for fluctuations in CDS spreads across sectors than the credit indices themselves. The additional information content over iTraxx seems to be related to some financial interest rates. We first use the estimated GRF to analyze the extent to which the eleven sectors we consider are systemic. After that, we use it to split the credit risk of individual firms into systemic, sectorial, and idiosyncratic components, and we perform some analyses to test that the estimated idiosyncratic components are actually firm-specific. The systemic and sectorial components explain around 65% of credit risk in the European industrial and financial sectors and 50% in the North American sectors, while 35% and 50% of risk, respectively, is of an idiosyncratic nature. Thus, there is a significant margin for portfolio diversification. We also show that our decomposition allows us to identify those firms whose credit would be harder to hedge. We end up analyzing the relationship between the estimated components of risk and some synthetic risk factors, in order to learn about the different nature of the credit risk components.

**Keywords:** systemic risk; sectorial risk; idiosyncratic risk; hedging; global risk factor

**JEL Classification:** C58; F34; G01; G32

## 1. Introduction

The turmoil in the financial systems around the world during the summer of 2007 brought up a strong debate on contagion. From the point of view of credit risk, financial regulators had focused on the capital to be required to each firm individually according to the probability of default of their debtors, as specified in BIS II (Bank for International Settlements, Basel II). However, the 2007 episode showed that the stand-alone risk of each company should not be the only characteristic to be considered. Rather, the contribution to the risk of the whole portfolio, or the possible contagion to the rest of firms should also be considered when determining capital requirements. It became clear that, to establish the appropriate framework for the prevention of financial crisis, it is crucial for financial supervisors to fully understand how contagion propagates throughout the firms in a sector and also among the different sectors (Ballester et al. 2016). Characterizing those firms and sectors that may have the greatest contagion effect on the rest is especially important, and it is also crucial to examine the

feedback contagion that may exist between the risk of default in some credit sectors, like the financial and the government (Acharya et al. 2014).

The financial crisis has also shown the importance of determining the main sources of risk in sovereign and corporate credit markets. This has become a requirement for financial institutions, since the Basel III agreement emphasizes that the credit strategy of a financial institution must take into account the cyclical aspects of the economy and the implied changes in the quality and composition of the overall credit portfolio. The credit strategy should be feasible in the long-term, through various economic cycles and shifting economic conditions, and financial institutions must know the sensitivity of their credit portfolio to a wide variety of macroeconomic and financial indicators.

Additionally, credit policies are required to ensure appropriate diversification at the portfolio level, and to have the ability to identify any particular sensitivities or concentrations (see BCBS 2000). For that, a central issue is to have an estimate of the interrelations among industry sectors and, in particular, the degree to which credit risk in a given sector has a systemic nature. Both issues: evaluating the systemic nature of sectorial credit risk, as well as the sensitivity of credit risk to changing economic and financial conditions, are examined in this paper.

We propose a simple methodology to estimate a global risk factor in credit markets from credit default swap CDS spreads, characterize its determinants and use it to decompose risk at the level of the firm into systemic, sectorial and idiosyncratic components. Indeed, the sensitivity of each firm to the global risk factor will suggest to what extent the risk in a credit portfolio is systemic. Firms with a higher systemic risk will be more prone to produce contagion across the credit market. We also show how to use the estimated global risk factor to estimate the sensitivity of credit portfolios to macroeconomic and financial indicators. Even though we use individual firms and sectorial indices to illustrate these important risk management applications, the same methods can be used for any credit portfolio.

The main lesson learnt from the crisis has shown that financial institutions need a comprehensive risk appetite framework in place that helps them to better understand and manage their risks by translating risk metrics and methods into strategic decisions, reporting, and day-to-day business decisions (FSB 2013) and (EBA 2014). Our analysis provides an element for such a risk appetite framework. By providing an estimate of the global risk factor, analyzing its determinants and using that factor to evaluate the systemic and the idiosyncratic components of risk, we describe an empirical framework that can be used by financial institutions to maintain the desired risk limits when taking their asset allocation decisions. Indeed, the sensitivity of credit risk from a particular sector or a geographic region to the global risk factor should help to take positions in anticipation of events affecting global risk and, in particular, to design an efficient hedge of a credit portfolio. Furthermore, by evaluating sectors with the most potential to produce systemic risk problems, our analysis should also be considered to be crucial for supervisors and regulators.

A sensible global credit risk factor could also be very useful when trying to anticipate the occurrence of a stress period in credit markets. Precisely, our analysis is a good starting point to evaluate credit risk exposures under stressful conditions, another requirement from Basel III. Stress testing requires identifying possible events or future changes in economic conditions that might have an adverse effect on a bank's credit exposures, as well as evaluating the bank's ability to withstand such changes. Banks should mainly examine (i) economic or industry downturns; (ii) market-risk events; and (iii) liquidity conditions. Banks should attempt to identify the type of situations, such as economic downturns, both in the whole economy and in particular sectors, and the combinations of credit and market events that might produce substantial losses. The most widely used measures of systemic risk are based on information on CDS spreads, which are forward-looking and reflect the market perception of the credit risk of the firm. This may be a more appropriate approach than using just accounting data. Indeed, the authors in (Gropp et al. 2006) show that market information on banking firms provided by their equity and bonds is a complementary information to their accounting data, and it should be relevant to analyze the nature of risk at financial institutions. In addition,

there is empirical evidence suggesting that CDS capture advanced information that is incorporated later on by rating agencies, which ends up being crucial to determine the level of required capital (Hull et al. 2004).

Regarding our specific contributions, we start by constructing sectorial credit indices from daily CDS spread data for individual firms from different regions in a given sector. This is a reasonable approach, since there is more similarity among CDS spreads from the same sector in different regions than among CDS spreads of different sectors in the same region. Then, we use a principal component analysis across sectorial indices to construct the global risk factor. The observed high commonality among sectorial indices suggests that this is a sensible characterization of a global risk factor. Our estimates are robust to alternative strategies for the estimation of sectorial credit indices. When we use our global risk factor to decompose credit risk at the level of the firm into systemic, sectorial and idiosyncratic components, we obtain relatively large idiosyncratic components of risk that are still larger in North American than in European firms, which may be due to a lack of liquidity. We provide evidence suggesting that portfolios made up of firms with higher idiosyncratic components are easier to hedge, contrary to what happens with portfolios made up of firms with lower idiosyncratic risk components. That is observed uniformly over the industrial and financial sectors of Europe and North America. We also show that well diversified CDS portfolios from a given sector have good possibilities for hedging by taking a contrary position in iTraxx or CDX indices or in their derivative products. Finally, we explore the nature of each estimated risk component by analyzing its sensitivity to some synthetic risk factors.

The paper is structured as follows: In the next section, we review the most relevant literature on this topic. In Section 3, we describe our database for CDS spreads and financial and macroeconomic indicators, construct sectorial credit indices, and examine their time evolution. In Section 4, we estimate the global risk factor and analyze in detail its information content. In Section 5, we use the global risk factor to decompose sectorial credit risk into systemic and idiosyncratic components, and we decompose credit risk at the level of the firm for the industrial and financial sectors of Europe and North America into systemic, sectorial and idiosyncratic components. In Section 6, we do an evaluation of the proposed methodology for risk decomposition, while Section 7 presents a robustness analysis of our approach for a global risk factor estimation. In Section 8, we examine the nature of each estimated risk component by analyzing their relationship with some synthetic risk factors. Finally, we conclude with a summary of the main findings.

## 2. Literature Review

Given the importance of the topic for researchers and market regulators after the financial crisis, the recent literature on measuring systemic risk has been quite extensive, and, in this section, we briefly review the papers we consider most relevant for our work.

A first strand of literature has considered the characterization of an indicator of systemic risk, with the principal component methodology playing a prominent role in that search. Using a sample of 150 European firms from January 2003 to July 2007, Berndt and Obreja (2010) show that the first principal component of CDS returns explained 46% of the variation in weekly CDS returns, even after correcting for a market factor (weekly excess return on the Morgan Stanley Capital International Europe (MSCI) index) Europe index) and a term premium, computed as the spread between the weekly return on the ten-year Euribor bond and the one-week Euribor. Chen and Härdle (2015) found that the first principal component for a set of eight iTraxx and CDX indices of 5- and 10-year maturities and investment grade and high-yield ratings explained 58.7% of the variance in the pre-crisis period, increasing up to 72.3% of the variance in the crisis period, but only 47% in the post-crisis period. They also concluded that a four-factor model could provide a good fit to weekly changes in CDS indices, with all factors receiving a significant market price. Bhansali et al. (2008) also use a three-jump model, and a different dataset from ours to carry out a decomposition of CDS spreads among systemic risk, sectorial risk and idiosyncratic risk. Duellmann and Masschelein (2007) used the analytic value-at-risk

approximation developed by Pykhtin (2004) which only requires risk parameters on a sector level. That approach is used to quantify the impact of credit concentrations in business sectors on the economic capital of credit portfolios.

Most of the literature on this type of decomposition has focused on the financial sector, at firm level. Rodríguez-Moreno and Peña (2013) analyzed two groups of systemic risk measures when searching for the best systemic indicator over the January 2004–November 2009 sample period. A first group contained indicators related to the overall tension in the market, while a second group was made up by indicators related to the contributions of individual institutions to systemic risk. In a sample of 20 European banks and 13 US banks they found that the first principal component of CDS spreads performed better as a systemic indicator than measures of market stress. Using daily data from 15 financial institutions from Europe and the US from January 2004 to June 2010, Giglio (2010) showed that the upturn in bond yields and CDS spreads of financial institutions during the crisis reflected increases in idiosyncratic default risk rather than systemic risk. This was the case for the months before the Bear Stearns episode on 15 March 2008, and also after Lehman's default. Hammoudeh et al. (2013) examined the behavior of the US 5-year sector CDS spread indices for banking, the financial services and the insurance sector over the period January 2004 to March 2009, suggesting the existence of an important systemic component of credit risk in the three sectors. Puzanova and Düllmann (2013) present an approach for measuring systemic risk and decomposing it into the contributions of individual institutions. To assess the system-wide loss, they modeled a banking sector as a portfolio comprising banks' net of capital liabilities, using a widely used credit risk model to assess the tail risk of such a portfolio. The model inputs were the banks' individual probabilities of default, the size of their net of capital liabilities and the banks' sensitivity to systemic factors, which capture correlations between banks' asset returns. Eder and Keiler (2015) estimated the degree of systemic risk and the magnitude of risk spillover effects by introducing a specific weighting scheme in a regression that relates observations to each other. They measure contagion effects in CDS levels as well as CDS changes. Their methodology allows for a decomposition of the total risk charge into a systemic, systemic and idiosyncratic risk charge. They found considerable spillovers of risk due to the interconnectedness of the systemically important banks and insurance companies in their sample. Depending on the state of the economy, up to a fifth of the total predicted CDS spread changes were due to financial contagion, emphasizing the need for macro-prudential supervision and providing an alternative explanation for the nonlinear relationship between a debtor's theoretical probability of default and observed credit spreads. The decomposition of risk into its systemic and idiosyncratic components has also been extensively studied for sovereign CDS markets, but we do not review it here. (Heitfield et al. 2006) examine the influence of systematic and idiosyncratic risk on credit losses for portfolios of large wholesale bank loans. They concluded that the relative importance of expected loss, systematic risk, and idiosyncratic risk varies considerably from sector-to-sector and is sensitive to the distribution of exposures within a given portfolio.

Other important contributions to the literature have examined the relevance of systemic risk in sovereign credit, using CDS spreads. Hilscher and Nosbusch (2010) studied the relative importance of country-specific and global factors on sovereign debt prices for a set of 31 emerging market countries from 1994 to 2007, to find that country-specific fundamentals have substantial explanatory power, even after controlling for global factors. Longstaff et al. (2011) found that sovereign credit risk tends to be much more correlated across countries than are equity index returns for the same countries. Their outcome suggested that the source of these higher correlations is the dependence of sovereign credit spreads on a common set of global market factors, risk premiums, and liquidity patterns. Badaoui et al. (2013) applied a factor model to decompose sovereign credit default swaps (CDS) spreads into default, liquidity, systemic liquidity and correlation components. Their analysis shows that sovereign CDS spreads were highly driven by liquidity, while sovereign bond spreads are less subject to liquidity frictions and therefore could represent a better proxy for sovereign default risk. Finally, their empirical results suggested that the increase in CDS spreads produced during the crisis was

mainly due to a surge in liquidity rather than to an increase in default intensity. Heinz and Sun (2014) found that European countries' sovereign CDS spreads are largely driven by global investor sentiment, macroeconomic fundamentals and liquidity conditions, even though the relative importance of these factors changed over time.

In terms of the determinants of risk, Berndt and Obreja (2010) and Chen and Härdle (2015) try to characterize the most influential financial variables that explain credit spread movements by analyzing the impact of some financial variables on individual CDS spreads and CDS indices, respectively. Much of this literature has focused on individual firm data, using accounting data and firm's characteristics as indicators of credit risk. Often, the goal has been to explain default rates. Our objective is somewhat different, as we use a wide set of macroeconomic and financial variables to explain the time evolution of corporate sector CDS indices according to the Industry Classification Benchmark. Schwaab et al. (2017) studied the dynamic properties of systemic default risk conditions for firms in different countries, industries and rating groups. They found that macro and default-specific world factors are a primary source of default clustering across countries. Defaults cluster more than implied by shared exposures to macro factors, suggesting that other factors also play a relevant role. Across firms, deviations of systemic default risk from macro fundamentals were correlated with net tightening bank lending standards, suggesting that bank credit supply and systemic default risk were inversely related.

## 3. A Historical Examination of Sectorial CDS Data

### 3.1. The Data

We use the database provided by Markit, the main supplier of CDS prices Markit (2008) and Markit (2012). Markit provides information on CDS spreads with different tenors: 6M, 1Y, 2Y, 3Y, 4Y, 5Y, 7Y, 10Y, 15Y, 20Y, and 30Y. The most liquid CDS is the 5-year contract. All these prices are composite, which means that for a given restructuring event, firm and currency, they are the average of prices provided by different financial institutions. The 'Sector' field is based on the ICB classification, (Industry Classification Benchmark), which distinguishes four levels: Industry, Supra Sector, Sector, and Subsector, and we work at Markit industry level, which considers 11 industries: energy, basic materials, industrials, consumer goods, consumer services, health care, financials, technology, telecommunication services, utilities, and government.[1] Finally, Markit identifies 13 different regions: Africa, Asia, Caribbean, Eastern Europe, Europe, India, Latin America, Middle East, North America, Oceania, OffShore, Pacific and Supranational.

We consider the data on 5-year CDS trading of senior unsecured debt, with 2608 daily observations between January 2006 and December 2015 on approximately 2500 firms from the 11 mentioned industries and the 13 geographical areas. We select the 760 firms having daily quotes on their 5-year CDS without having been subject to a merger or acquisition. Most of the CDSs have ratings "BBB" or "A". The best represented sectors are financials, consumer services and industrials, while the main regions are North America, Europe, and Asia. These distributions are relatively stable over time.

To derive a fundamental interpretation of some of the estimates we compute throughout the paper, we will use a wide set of daily indicators from the Bloomberg database. For the purpose of interpreting our results, we classify them as pure financial indicators, equity indicators, risk aversion indicators and indicators bearing some relationship with macroeconomic or monetary policy. Some of the relationships of credit spreads with financial market indicators may be short-lived, and they may be lost if we aggregate to monthly frequencies. It would clearly be interesting to analyze the relationship with business cycle indicators, but our interest here is to evaluate what type of indicators have a stronger influence on credit spreads.

---

[1]　Government is a category considered by Markit but not included in the Industry Classification Benchmark.

Financial indicators: (1) three-month EURIBOR interest rate, (2) three-month EONIA rate, (3) three-month USD LIBOR Interest Rate, (4) three-month overnight index swap (OIS), (5) one-year EURO Swap Rate, (6) five-year EURO Swap Rate, (7) ten-year EURO Swap Rate, (8) one-year USD Swap Rate, (9) five-year USD Swap Rate, (10) ten-year USD Swap Rate, (11) one-year JPY Swap rate, (12) five-year JPY Swap rate, and (13) ten-year JPY Swap rate. We use the OIS index and the EONIA rate just to construct liquidity indicators, as explained below.

Equity Indicators (the 10 MSCI global equity indices) (14)–(23): MSCI World/Basic materials, MSCI World/Consumer goods, MSCI World/Consumer services, MSCI World/Energy, MSCI World/Financials, MSCI World/Healthcare, MSCI World/Industrials, MSCI World/Technology, MSCI World/Telecommunication services and MSCI World/Utility.

Risk aversion indicators: (24) USD liquidity risk premium, measured by the absolute difference between three-month LIBOR and the three-month OIS Index. (25) Euro liquidity premium, measured by the absolute difference between three-month EURIBOR and three-month EONIA, both in euros, (26) three-month five-year USD swaption, (27) three-month five-year Euro swaption, (28) the VIX Volatility Index, from CBOE, as market expectations of near-term volatility conveyed by S&P 500 stock index option prices, (29) the VSTOXX Index, as implied volatility in EURO STOXX 50 real-time option prices, (30) implied volatility from option prices for the three-month euro-dollar exchange rate, (31) implied volatility from the 3-month ATM iTraxx Europe Index options, and (32) implied volatility from the 3-month ATM CDX North American Investment Grade Index Option.[2]

The term Libor–OIS spread is considered to be a measure of the health of banks because it reflects their views on the risk of default associated with lending to other banks. Indeed, former Fed Chairman Alan Greenspan stated recently that the "Libor–OIS remains a barometer of fears of bank insolvency" (see Thornton et al. 2009). We use indicators (24)–(25) as measures of liquidity-premium which may contain information about stress in the money markets (see Beirne 2012; Blix Grimaldi 2010). The implied volatility of interest rates in indicators (26)–(27) captures the market uncertainty on future monetary policy.

Derivative prices contain information about the probability assessment by market participants of the outcome of the underlying asset price upon maturity. Such an information can be extracted using risk-neutral density functions that can provide a forward-looking insight into the risk sentiment of market participants. Indeed, the difference between the risk-neutral and utility-adjusted density function yields a measure of relative risk aversion of the representative investor. Options are forward looking in nature and thus are a useful source of information for judging market sentiment about future prices of financial assets and their dynamics as it has been stressed elsewhere (see Siegel 1997; Campa and Chang 1998; Lopez and Walter 2000; Skintzi and Refenes 2005; Driessen et al. 2009; Ornelas 2019; Hui et al. 2013 among any others). In particular, the VIX and VSTOXX indices are commonly treated as quick and easy proxies for risk appetite, and they are primarily designed to measure market expectations of volatility in the equity market. They are derived from S&P 500 and Euro Stoxx 50 options implied volatilities, which investors buy and sell to change the amount of risk to which they are exposed (see Illing and Aaron 2005; Blix Grimaldi 2010). Thus, we use implied volatility variables (28)–(32) as measures of risk aversion for different assets: interest rates, equity, forex, or credit.

Macroeconomic indicators: (33) euro-dollar exchange rate, (34) dollar-yen exchange rate, (35) 5-year German government yield, (36) 10-year German government bond yield, (37) 5-year US Treasury Rate, (38) 10-year US Treasury yield, (39) the 10-year yield on Japan government debt, (40)–(42) term structure slope, defined as the 10-year, 1-year rates spread, $r_{t,10} - r_{t,1}$, in swap rates in

---

2    The implicit volatilities from 3-month ATM iTraxx Europe Index Option and the 3-month ATM CDX North American Investment Grade Index Option were provided by JP Morgan.

US dollar, euro, and yen, and (44)–(45) term structure curvature, defined as $r_{t,10} - 2r_{t,5} + r_{t,1}$, from swap rates in US dollar, euro and yen.

Credit indicators: (46) iTraxx Europe IG index, (47) CDX North America IG index, (48) ITraxx Japan IG index, (49) ITraxx Europe Hi Vol index, (50) CDX North America HY index.

Table 1 shows the main statistics for the set of the economic and financial indicators, while Table 2 shows the values of the same statistics for the first difference of the indicators. All of them can be seen to be stationary in first differences.

**Table 1.** Economic and financial indicators: main statistics.

| Variable | Mean | Median | St. Dev. | Minimum | Maximum | DFA(1) |
|---|---|---|---|---|---|---|
| Euribor 3m | 1.64 | 0.91 | 1.67 | −0.19 | 5.38 | −0.70 |
| Eonia 3m | 1.31 | 0.47 | 1.57 | −0.31 | 4.34 | −0.54 |
| USD Libor 3m | 1.60 | 0.38 | 1.99 | 0.22 | 5.71 | −1.57 |
| USD OIS 3m | 1.29 | 0.17 | 1.97 | 0.07 | 5.41 | −1.74 |
| Euro swap 1y | 1.84 | 1.26 | 1.65 | −0.13 | 5.42 | −0.21 |
| Euro swap 5y | 2.35 | 2.35 | 1.45 | 0.11 | 5.15 | 0.02 |
| Euro swap 10y | 2.89 | 3.16 | 1.26 | 0.46 | 5.07 | 0.15 |
| USD swap 1y | 1.67 | 0.54 | 1.91 | 0.26 | 5.72 | −1.63 |
| USD swap 5y | 2.62 | 1.98 | 1.48 | 0.75 | 5.72 | −1.28 |
| USD swap 10y | 3.36 | 3.04 | 1.21 | 1.57 | 5.82 | −1.13 |
| Japan swap 1y | 0.47 | 0.36 | 0.29 | 0.00 | 1.14 | −0.49 |
| Japan swap 5y | 0.74 | 0.58 | 0.45 | 0.00 | 1.70 | −0.53 |
| Japan swap 10y | 1.21 | 1.18 | 0.48 | 0.24 | 2.23 | −0.48 |
| MSCI World/Basic materials | 229.65 | 228.91 | 38.35 | 119.01 | 334.81 | −2.34 |
| MSCI World/Consumer goods | 146.98 | 136.43 | 34.13 | 85.78 | 211.15 | −0.44 |
| MSCI World/Consumer services | 128.28 | 119.94 | 36.86 | 55.59 | 202.76 | −0.48 |
| MSCI World/Energy | 239.56 | 238.48 | 34.32 | 148.49 | 335.24 | −2.34 |
| MSCI World/Financial | 99.80 | 95.06 | 30.03 | 38.44 | 166.56 | −1.33 |
| MSCI World/Healthcare | 129.83 | 113.95 | 39.74 | 72.08 | 227.38 | −0.00 |
| MSCI World/Industrials, | 158.21 | 157.77 | 31.09 | 72.59 | 208.88 | −1.41 |
| MSCI World/Technology | 96.46 | 90.90 | 24.33 | 47.40 | 150.62 | −0.50 |
| MSCI World/Telecommunication services | 61.50 | 60.50 | 8.68 | 41.26 | 81.52 | −1.73 |
| MSCI World/Utility | 117.50 | 111.91 | 17.50 | 85.65 | 168.79 | −1.70 |
| Liquidity USD Premium (USD Libor 3m-USD OIS 3m) | 0.31 | 0.15 | 0.40 | 0.05 | 3.43 | −4.28 |
| Liquidity Euro premium (Euribor 3m-Eonia 3m) | 0.33 | 0.20 | 0.33 | 0.05 | 1.85 | −2.81 |
| USD Swaption 3m5y | 94.16 | 81.81 | 35.97 | 41.64 | 208.15 | −2.34 |
| Euro Swaption 3m5y | 69.93 | 66.36 | 25.35 | 26.92 | 166.75 | −2.41 |
| VIX Index | 20.39 | 17.38 | 9.69 | 10.18 | 72.92 | −3.14 |
| VSTOXX | 24.53 | 22.45 | 9.16 | 13.22 | 71.49 | −3.69 |
| Imp Vol 3m EURUSD fx | 10.30 | 10.08 | 3.43 | 4.90 | 23.56 | −2.05 |
| Imp Vol 3m Itraxx IG | 0.64 | 0.60 | 0.24 | 0.00 | 1.44 | −3.60 |
| Imp Vol 3m CDX IG | 0.58 | 0.53 | 0.21 | 0.00 | 1.15 | −3.42 |
| EURUSD fx | 1.33 | 1.33 | 0.11 | 1.06 | 1.59 | −1.81 |
| JPYUSD Fx | 0.01 | 0.01 | 0.00 | 0.01 | 0.01 | −1.12 |
| German Government Bond 5y | 1.90 | 1.78 | 1.48 | −0.25 | 4.70 | −0.33 |
| German Government Bond 10Y | 2.57 | 2.76 | 1.25 | 0.09 | 4.64 | −0.33 |
| US Treasury bond 5y | 2.26 | 1.73 | 1.32 | 0.58 | 5.18 | −1.49 |
| US Treasury bond 10y | 3.10 | 2.90 | 1.03 | 1.44 | 5.21 | −1.49 |
| Japan Government Bond 10y | 1.10 | 1.15 | 0.47 | 0.07 | 1.98 | −0.30 |
| US slope | 1.69 | 1.71 | 0.99 | −0.25 | 3.34 | −1.82 |
| EUR slope | 1.05 | 1.11 | 0.72 | −0.48 | 2.39 | −1.42 |
| Yen slope | 0.75 | 0.69 | 0.30 | 0.23 | 1.77 | −2.19 |
| US curvature | −0.21 | −0.18 | 0.51 | −1.50 | 0.56 | −2.11 |
| EUR curvature | 0.04 | 0.14 | 0.35 | −0.91 | 0.73 | −1.96 |
| Yen curvature | 0.21 | 0.27 | 0.24 | −0.64 | 0.49 | −1.94 |
| Itraxx IG | 90.96 | 89.07 | 44.19 | 20.56 | 204.24 | −2.11 |
| CDX IG | 90.96 | 85.41 | 42.17 | 29.96 | 264.60 | −2.33 |
| Itraxx Japan IG | 111.37 | 96.92 | 81.73 | 14.75 | 542.98 | −2.61 |
| HiVol Itraxx | 143.62 | 136.76 | 87.63 | 39.69 | 500.58 | −2.00 |
| CDX HY | 522.60 | 447.84 | 264.75 | 188.54 | 1775.51 | −2.13 |

Note: The table shows the mean, median, standard deviation, minimum and maximum of each indicator. The last column is the Dickey–Fuller statistic, including one lag of the dependent variable in the regression equation to estimate the Dickey–Fuller statistic.

**Table 2.** Economic and financial indicators (in differences): main statistics.

| Variable | Mean | Median | St. Dev. | Minimum | Maximum | DFA(1) |
|---|---|---|---|---|---|---|
| Euribor 3m | −0.005 | −0.001 | 0.05 | −0.33 | 0.21 | −7.94 |
| Eonia 3m | −0.005 | 0.000 | 0.05 | −0.40 | 0.09 | −7.90 |
| USD Libor 3m | −0.008 | 0.000 | 0.09 | −0.87 | 0.64 | −11.23 |
| USD OIS 3m | −0.008 | 0.000 | 0.05 | −0.55 | 0.09 | −7.85 |
| USD swap 1y | −0.008 | −0.002 | 0.07 | −0.39 | 0.42 | −12.22 |
| USD swap 5y | −0.007 | −0.012 | 0.10 | −0.40 | 0.48 | −13.29 |
| USD swap 10y | −0.006 | −0.008 | 0.11 | −0.53 | 0.38 | −13.39 |
| Euro swap 1y | −0.006 | −0.002 | 0.06 | −0.27 | 0.21 | −9.78 |
| Euro swap 5y | −0.006 | −0.007 | 0.07 | −0.29 | 0.30 | −14.00 |
| Euro swap 10y | −0.005 | −0.009 | 0.08 | −0.39 | 0.28 | −14.95 |
| Japan swap 1y | 0.000 | 0.000 | 0.02 | −0.11 | 0.10 | −13.16 |
| Japan swap 5y | −0.002 | −0.003 | 0.04 | −0.18 | 0.15 | −14.05 |
| Japan swap 10y | −0.003 | −0.005 | 0.04 | −0.20 | 0.15 | −14.77 |
| MSCI World/Basic materials | −0.016 | 0.442 | 6.68 | −36.80 | 18.56 | −15.36 |
| MSCI World/Consumer goods | 0.200 | 0.403 | 2.02 | −12.82 | 5.73 | −15.07 |
| MSCI World/Consumer services | 0.144 | 0.534 | 2.49 | −12.81 | 7.28 | −14.66 |
| MSCI World/Energy | −0.072 | 0.580 | 6.62 | −37.95 | 17.50 | −15.77 |
| MSCI World/Financial | −0.088 | 0.272 | 2.56 | −15.47 | 8.40 | −15.20 |
| MSCI World/Healthcare | 0.171 | 0.306 | 2.22 | −16.14 | 8.83 | −15.82 |
| MSCI World/Industrials, | 0.087 | 0.520 | 3.32 | −17.96 | 9.15 | −14.93 |
| MSCI World/Technology | 0.110 | 0.354 | 1.97 | −9.38 | 5.74 | −14.35 |
| MSCI World/Telecommunication services | 0.028 | 0.140 | 1.06 | −5.93 | 3.31 | −15.01 |
| MSCI World/Utility | 0.010 | 0.272 | 2.15 | −17.15 | 6.10 | −16.19 |
| Liquidity USD Premium (USD Libor 3m-USD OIS 3m) | 0.000 | 0.000 | 0.08 | −0.75 | 0.88 | −11.59 |
| Liquidity Euro premium (Euribor 3m-Eonia 3m) | 0.000 | −0.001 | 0.05 | −0.26 | 0.49 | −12.21 |
| USD Swaption 3m5y | −0.009 | −0.288 | 6.07 | −24.49 | 34.00 | −14.54 |
| Euro Swaption 3m5y | −0.044 | −0.140 | 4.22 | −24.19 | 24.90 | −16.80 |
| VIX Index | 0.017 | −0.122 | 2.65 | −14.77 | 16.16 | −15.98 |
| VSTOXX | 0.027 | −0.117 | 2.93 | −16.02 | 19.20 | −16.45 |
| Vol Imp 3m EURUSD fx | 0.002 | −0.021 | 0.63 | −2.52 | 4.98 | −16.47 |
| Vol Imp 3m Itraxx IG | 0.001 | −0.002 | 0.05 | −0.23 | 0.38 | −15.10 |
| Vol Imp 3m CDX IG | 0.001 | −0.001 | 0.04 | −0.15 | 0.43 | −14.32 |
| EURUSD fx | 0.000 | 0.000 | 0.02 | −0.06 | 0.09 | −14.17 |
| JPYUSD Fx | 0.000 | 0.000 | 0.00 | 0.00 | 0.00 | −13.63 |
| German Government Bond 5y | −0.006 | −0.005 | 0.09 | −0.34 | 0.25 | −14.51 |
| German Government Bond 10Y | −0.005 | −0.009 | 0.08 | −0.30 | 0.31 | −15.27 |
| US Treasury bond 5y | −0.006 | −0.008 | 0.10 | −0.40 | 0.35 | −13.19 |
| US Treasury bond 10y | −0.004 | −0.009 | 0.10 | −0.41 | 0.33 | −13.38 |
| Japan Government Bond 10y | −0.003 | −0.004 | 0.04 | −0.21 | 0.14 | −15.04 |
| US slope | 0.002 | −0.005 | 0.10 | −0.48 | 0.34 | −13.61 |
| EUR slope | 0.000 | −0.005 | 0.07 | −0.22 | 0.36 | −14.34 |
| Yen slope | −0.002 | −0.004 | 0.04 | −0.18 | 0.14 | −15.05 |
| US curvature | 0.000 | 0.005 | 0.08 | −0.28 | 0.28 | −13.71 |
| EUR curvature | 0.001 | 0.002 | 0.05 | −0.24 | 0.18 | −15.07 |
| Yen curvature | 0.001 | 0.000 | 0.03 | −0.15 | 0.12 | −14.00 |
| Itraxx IG | 0.106 | −0.128 | 6.88 | −22.91 | 30.27 | −15.15 |
| CDX IG | 0.104 | −0.215 | 6.83 | −41.11 | 41.66 | −15.78 |
| Itraxx Japan IG | 0.122 | −0.195 | 13.50 | −93.52 | 66.92 | −14.76 |
| HiVol Itraxx | 0.076 | −0.331 | 12.31 | −61.54 | 87.86 | −14.72 |
| CDX HY | 0.003 | −0.016 | 0.37 | −2.14 | 2.48 | −14.35 |

Note: The table shows the mean, median, standard deviation, minimum and maximum of the first difference of each indicator. The last column is the Dickey–Fuller statistic, including one lag of the dependent variable in the regression equation to estimate the Dickey–Fuller statistic.

### 3.2. Sectorial CDS Indices

### 3.2.1. Main Statistics

We construct daily CDS indices for each sector by taking the median CDS spread traded each day in that sector across all firms in all regions, with the results shown in Figure 1. Weekly sectorial data are obtained by taking weekly averages of the daily observations for each sectorial credit index. Finally, we compute logarithmic changes of weekly CDS spreads, obtaining a total of 365 weekly

observations over the 2006–2016 period. This is the sectorial credit data we use in what follows. Their main statistical characteristics are displayed in Table 3. Sectorial CDS indices are clearly non-stationary, while their weekly changes are stationary, as it can be confirmed from the application of Dickey–Fuller tests (rows 10–15 in Table 3). The higher volatility is achieved by weekly changes in spreads from telecommunication services and the government sector. Interestingly enough, all sectors display right skewness, while kurtosis is particularly high in the financial, government, health care and utilities sectors. As a consequence, the assumption of normality as the distribution of weekly changes in CDS spreads is overwhelmingly rejected in all sectors.

**Table 3.** Sectorial returns. Main statistics.

| Statistic | BM | CG | CS | EN | FIN | GOV | HC | IND | TEC | TEL | UTI |
|---|---|---|---|---|---|---|---|---|---|---|---|
| Mean*100 | 0.171 | 0.106 | 0.135 | 0.303 | 0.304 | 0.282 | 0.087 | 0.150 | −0.008 | 0.174 | 0.168 |
| Standard deviation*100 | 4.15 | 3.82 | 3.92 | 4.29 | 4.57 | 5.42 | 3.96 | 4.02 | 4.09 | 5.13 | 3.46 |
| Volatility | 29.9% | 27.5% | 28.3% | 30.9% | 32.9% | 39.1% | 28.6% | 29.0% | 29.5% | 37.0% | 24.9% |
| Skewness | 0.69 | 0.71 | 0.38 | 0.85 | 1.65 | 1.18 | 0.64 | 0.78 | 0.29 | 0.67 | 1.29 |
| Kurtosis | 5.43 | 6.88 | 4.96 | 6.73 | 11.84 | 9.28 | 9.52 | 6.36 | 3.86 | 5.31 | 8.91 |
| Maximum | 0.20 | 0.19 | 0.18 | 0.24 | 0.30 | 0.34 | 0.23 | 0.19 | 0.16 | 0.20 | 0.19 |
| minimum | −0.14 | −0.16 | −0.15 | −0.18 | −0.16 | −0.22 | −0.21 | −0.15 | −0.11 | −0.19 | −0.11 |
| Range | 0.34 | 0.35 | 0.33 | 0.42 | 0.46 | 0.56 | 0.43 | 0.34 | 0.28 | 0.39 | 0.31 |
| Jarque–Bera | 168.7 | 369.8 | 96.0 | 363.9 | 1933.4 | 978.0 | 958.2 | 297.0 | 23.5 | 154.0 | 902.7 |
| **Unit Root Tests** | | | | | | | | | | | |
| Index levels | | | | | | | | | | | |
| adf1 | −2.09 | −1.97 | −1.90 | −1.91 | −1.92 | −2.06 | −2.18 | −1.83 | −1.83 | −2.38 | −1.76 |
| adf4 | −2.28 | −2.25 | −2.20 | −2.11 | −2.16 | −2.18 | −2.37 | −2.05 | −1.87 | −2.34 | −2.09 |
| adf8 | −2.38 | −2.08 | −2.29 | −2.69 | −2.23 | −2.54 | −2.42 | −2.40 | −1.92 | −2.82 | −2.01 |
| Weekly changes | | | | | | | | | | | |
| adf1 | −12.18 | −13.57 | −13.85 | −12.35 | −11.32 | −13.86 | −15.45 | −12.88 | −14.80 | −14.50 | −12.15 |
| adf4 | −8.65 | −8.66 | −8.88 | −7.46 | −8.74 | −8.71 | −8.23 | −8.23 | −9.18 | −10.12 | −8.36 |
| adf8 | −6.98 | −6.98 | −6.84 | −5.76 | −7.06 | −6.52 | −6.80 | −6.45 | −7.58 | −6.95 | −7.10 |

Note: Main statistics of weekly changes in sectorial indices. The Augmented Dickey–Fuller (ADF) statistics for the sectorial indices in levels are shown in the upper panel under 'Unit root tests', while Augmented Dickey–Fuller for weekly changes of sectorial indices are shown in the lower panel. Both are implemented with one lag of the dependent variable. Critical value at 95% for the ADF test is −2.87. BM = Basic materials, CG = Consumer goods, CS = Consumer services, EN = Energy, FIN = Financials, GOV = Government, HC = Health care, IND = Industrials, TEC = Technology, TEL = Telecommunication services, and UTI = Utilities.

### 3.2.2. Time Evolution

The time evolution of the sectorial indices shows the main market events that took place during the sample period. New Century Financial, largest U.S. subprime lender, filed for Chapter 11 bankruptcy, announcing the departure of more than half the workforce as of 2 April 2007, with no apparent impact on CDS spreads. Even more surprisingly, CDS spreads barely increased at the time of the Bear Sterns crisis in July 2007. However, the market was accumulating fears on the situation of credit. The origin of the financial crisis may be placed on 9 August 2007, with BNP Paribas announcing that it was ceasing activity in three hedge funds that specialized in US mortgage debt. The announcement acted as a signal that there were tens of trillions of US dollar worth of derivatives which were worth much less than previously estimated. Since nobody knew the exposure of individual banks to these toxic assets, trust evaporated overnight and banks stopped doing business with each other. The perception of risk spread over all sectors, which explains the simultaneous increase shown in Figure 1 Spreads peaked in March 2008 at a median level of 150 bp. After a temporary reduction, spreads again started a moderate increasing trend, well before the difficulties that would come later in the summer. The Lehman Brothers bankruptcy brought them to a median level across sectors of 300 bp. by March 2009. The anticipation of an impending crisis in CDS spreads over 2008 has not been sufficiently emphasized in the credit risk literature.

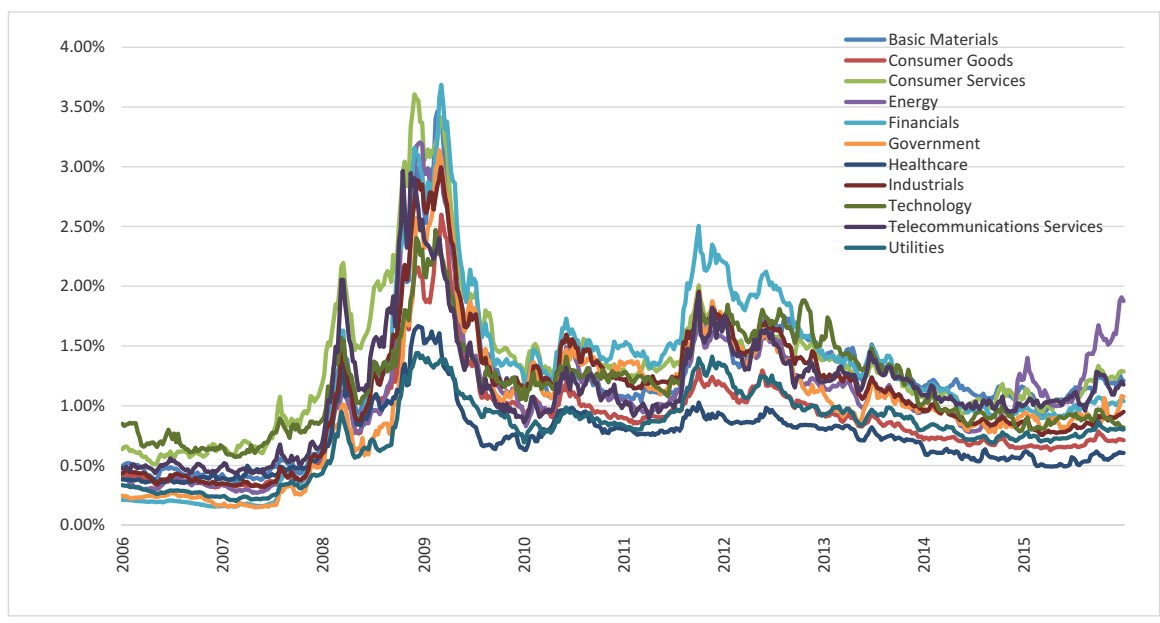

**Figure 1.** Sectorial CDS spreads 2006–2015. The figure shows our estimated sectorial CDS indices. Weekly data: January 2006–December 2015.

CDS spreads reached their highest values during the first quarter of 2009. Finally, the American Recovery and Reinvestment Act of 2009, commonly referred to as the Stimulus or Recovery Act, was an economic stimulus package enacted by the 111th United States Congress in February 2009 and signed into law on 17 February 2009 by President Barack Obama. At the London G20 on 2 April 2009, world leaders committed themselves to a $5 trillion (£3 tn) fiscal expansion, an extra $1.1 trillion in resources to help the International Monetary Fund and other global institutions boost jobs and growth, and to reform the banks. During this period of government stimulus measures, spreads rapidly decreased, although they stayed well above their pre-crisis levels.

The S&P downgrade of US sovereign debt in 5 August 2011 again brought a sharp increase in CDS spreads. In September, rating agencies downgraded the sovereign debt of some European countries, and the U.S. Treasury Secretary urged European officials to deal with the crisis and avoid "catastrophic risks" after flying to a meeting of European Union finance chiefs in Poland. The European debt crisis produced another rise in CDS spreads. Only the decisive intervention of the ECB president in July 2012 could make CDS spreads finally start a slow reduction. At the end of 2015, the median across sectors was 105 bp., versus the median of 40 bp. at the beginning of 2006.

Figure 1 shows that, over the whole sample, sectorial indices show a good deal of low frequency co-movement, with fluctuations of different sizes across sectors. Additionally, linear correlation coefficients among percent changes in CDS spreads in Table 4 show significant high-frequency co-movements across all sectors. Median correlation coefficients for each sector (last row in the table) are around 0.70, except for health care and technology, which have a median correlation in the neighborhood of 0.50 suggesting that they might be the less systemic sectors. The high overall correlations reflect the existence of at least a common factor, while the lower association between the health care and technology sectors and all the others must be due to the existence of specific factors explaining fluctuations in CDS prices in these two sectors. On the other hand, no single sector seems to be dominant, in the sense of having higher correlations with all the other sectors.

**Table 4.** Sectorial correlation matrix.

| Sector | BM | CG | CS | EN | FIN | GOV | HC | IND | TEC | TEL | UTI |
|--------|------|------|------|------|------|------|------|------|------|------|------|
| BM | 100% | 73% | 67% | 67% | 68% | 61% | 47% | 76% | 54% | 62% | 64% |
| CG | 73% | 100% | 71% | 69% | 77% | 65% | 49% | 80% | 56% | 67% | 73% |
| CS | 67% | 71% | 100% | 65% | 66% | 57% | 44% | 71% | 47% | 66% | 64% |
| EN | 67% | 69% | 65% | 100% | 73% | 62% | 46% | 70% | 51% | 64% | 71% |
| FIN | 68% | 77% | 66% | 73% | 100% | 72% | 49% | 77% | 53% | 67% | 76% |
| GOV | 61% | 65% | 57% | 62% | 72% | 100% | 35% | 67% | 47% | 56% | 65% |
| HC | 47% | 49% | 44% | 46% | 49% | 35% | 100% | 52% | 35% | 42% | 48% |
| IND | 76% | 80% | 71% | 70% | 77% | 67% | 52% | 100% | 55% | 67% | 72% |
| TEC | 54% | 56% | 47% | 51% | 53% | 47% | 35% | 55% | 100% | 50% | 51% |
| TEL | 62% | 67% | 66% | 64% | 67% | 56% | 42% | 67% | 50% | 100% | 66% |
| UTI | 64% | 73% | 64% | 71% | 76% | 65% | 48% | 72% | 51% | 66% | 100% |
| **Median** | **67%** | **71%** | **66%** | **67%** | **72%** | **62%** | **47%** | **71%** | **51%** | **66%** | **66%** |

Note: Pairwise correlation matrix between weekly changes in sectorial indices. BM = Basic materials, CG = Consumer goods, CS = Consumer services, EN = Energy, FIN = Financials, GOV = Government, HC = Health care, IND = Industrials, TEC = Technology, TEL = Telecommunication services, UTI = Utilities, and MCS = Median intra correlation for each sector.

### 3.2.3. The Relationship of Sectorial CDS Indices with Credit and Equity Market Indices

To evaluate the characteristics of the type of risk involved in a sectorial credit position, we start by showing, in Table 5 least-squares, single equation estimates of regressions of the sectorial indices on iTraxx, a natural choice as a risk factor:

$$I_t = \beta_0 + \beta_1 iTraxx + u_t, \tag{1}$$

where $I_t$ represents each sectorial index. As we can see, estimated betas lie in the (0.27; 0.51) interval, with R-squared values (column 3) ranging from just above 0.20 for health care and technology, the two sectors with lower correlations with the rest, to the neighborhood of 0.50 for the consumer goods, telecommunication services and the financial sectors. Since our sample includes European and North American firms, it is not surprising that the CDX IG Index also has a noticeable explanatory power (not shown in the table). Test statistics in the table show that regression residuals are not normally distributed, and display significant autocorrelation and Autoregressive Conditional Heteroskedasticity (ARCH) effects in most sectors. That reduces the precision of least-squares estimates, creating some difficulties for hypothesis testing. However, we are not interested in running any inferential analysis on these estimates, but, instead, on examining the explanatory power of iTraxx as a factor explaining sectorial credit indices. To that end, stationary residuals, as suggested by the Augmented Dickey–Fuller (ADF) statistic test statistics in the last column, are crucial.

An important question for risk management would relate to the performance of a hedging strategy for a CDS position in a given sector, based on taking a contrary position in the iTraxx Index, using as hedge ratio the least-squares estimate of beta for that sector. Except for a constant, the residuals from these regressions would be the returns on the hedged portfolio, and the R-squared statistics show the reduction achieved by the hedge on the variance of the sector portfolio. With the exception of health care and technology sectors hedging efficiency would be significant, with a substantial reduction in sectorial credit index variance, between 32% and 53%, which shows an interesting potential for hedging credit portfolios when they are sufficiently diversified in a given sector.

It is also interesting to see in Table 6 that the MSCI stock indices contain significant information on sectorial credit indices (column 2),

$$I_t = \beta_0 + \sum_{i=1}^{11} \beta_i MSCI_{it} + u_t, \tag{2}$$

with a median R-squared of 0.34, showing that credit spreads react to events in the stock markets. When we add the iTraxx index to the set of sectorial MSCI indices (column 3),

$$I_t = \beta_0 + \sum_{i=1}^{11} \beta_i MSCI_{it} + \gamma iTraxx_t + u_t, \tag{3}$$

a comparison of its R-squared statistic with that of regresion (1) suggests that there is some information in MSCI indices on the credit market that is not captured by iTraxx.

**Table 5.** One-factor Itraxx regressions explaining sectorial credit indices.

| Factor: iTraxx | Beta | adj R2 | $\sigma$*100 | Jarque–Bera | LBQ(1) | LBQ(4) | LBQ(12) | Arch Test | ADF(1) |
|---|---|---|---|---|---|---|---|---|---|
| BM | 0.35 | 0.34 | 3.37 | 228.5 | 9.9 | 20.5 | 28.2 | 8.6 | −13.8 |
| CG | 0.38 | 0.47 | 2.78 | 207.1 | 25.8 | 27.9 | 45.0 | 43.6 | −15.2 |
| CS | 0.35 | 0.39 | 3.07 | 92.7 | 6.3 | 7.0 | 16.0 | 2.6 | −15.7 |
| EN | 0.38 | 0.37 | 3.41 | 264.7 | 59.2 | 79.6 | 98.4 | 90.4 | −13.0 |
| FIN | 0.48 | 0.53 | 3.13 | 2394.8 | 85.9 | 138.5 | 142.4 | 165.6 | −11.3 |
| GOV | 0.44 | 0.32 | 4.49 | 1089.4 | 12.7 | 15.3 | 22.1 | 19.4 | −14.9 |
| HC | 0.27 | 0.22 | 3.51 | 369.3 | 2.2 | 6.2 | 18.0 | 0.9 | −16.9 |
| IND | 0.39 | 0.44 | 3.01 | 276.8 | 17.7 | 20.7 | 45.3 | 24.3 | −13.9 |
| TEC | 0.29 | 0.24 | 3.56 | 38.5 | 0.1 | 3.4 | 10.6 | 28.2 | −16.3 |
| TEL | 0.51 | 0.47 | 3.75 | 34.7 | 6.8 | 13.9 | 22.2 | 14.9 | −16.0 |
| UTI | 0.33 | 0.44 | 2.58 | 340.2 | 25.9 | 46.4 | 57.8 | 18.4 | −14.0 |

Note: The table shows slope estimates, R2 statistics and the standard deviation of residuals in a least-squares regression of each sectorial index on iTraxx as the single explanatory variable. All regressions are estimated on weekly differences of both variables. The remaining columns show statistics for residual analysis. Jarque–Bera denotes the Normality statistic of that name, LBQ denotes the Ljung–Box Q statistic to test for residual autocorrelation, applied at at 1, 4 and 12 lags, Arch test denotes a test for first-order ARCH dependence, through the application of the LBQ test to the squared residuals, and ADF(1) is the Augmented Dickey–Fuller statistic including one lag of the dependent variable in the unit root regression. Critical values at 95% are: 5.86 for Jarque–Bera test; 3.84, 9.49 and 21.03 for Ljung–Box test at 1, 4 and 12 lags, respectively, 3.84 for the Arch test and –2.87 for the ADF test.

**Table 6.** Adjusted R2 statistics from alternative regressions explaining sectorial indices.

| Sector | MSCI | MSCI + iTraxx | MSCI + GRF | All Indicators |
|---|---|---|---|---|
| BM | 0.30 | 0.38 | 0.70 | 0.52 |
| CG | 0.36 | 0.50 | 0.77 | 0.65 |
| CS | 0.34 | 0.43 | 0.65 | 0.53 |
| EN | 0.37 | 0.43 | 0.70 | 0.53 |
| FIN | 0.43 | 0.57 | 0.79 | 0.72 |
| GOV | 0.28 | 0.35 | 0.65 | 0.43 |
| HC | 0.17 | 0.24 | 0.36 | 0.28 |
| IND | 0.36 | 0.48 | 0.79 | 0.59 |
| TEC | 0.19 | 0.25 | 0.43 | 0.31 |
| TEL | 0.39 | 0.51 | 0.67 | 0.56 |
| UTI | 0.31 | 0.46 | 0.70 | 0.55 |

Note: The table shows adjusted R2 statistics from running alternative least-squares regressions of each sectorial credit index, on a different set of explanatory variables. All regressions are estimated in weekly differences of the dependent variable and the regressors. The second column uses the 10 MSI sectorial indices, column 3 adds iTraxx as an additional explanatory variable. Column 4 uses the MSCI indices together with the Global Risk Factor (GRF), and column 5 uses all the indicators described in the data section, except for the credit indicators.

## 4. Estimating a Global Risk Factor

### 4.1. Estimation Methodology

We capture the commonality of risk across the different sectors by applying a principal component methodology to the set of time series of weekly changes in the sectorial credit indices. A similar strategy,

in the context of extensive commonality, has been used by (Ballester et al. 2016), among many others. If we denote by $X$ the $T$x11 matrix made up by the standardized weekly changes in the sectorial indices, the principal component analysis is based on the eigenvalues and eigenvectors of the correlation matrix $V = X'X/T$, of order $11 \times 11$. Let $W$ be the $11 \times 11$ orthogonal matrix having as columns the eigenvectors of $V$. The spectral decomposition of $V$ is: $V = W\Lambda W'$, with $\Lambda$ being the diagonal matrix of the eigenvalues of $V$, where we have ordered the columns in $W$ and the eigenvalues in $\Lambda$ accordingly. The principal components of $X$ are obtained as:

$$PC = XW, \tag{4}$$

the product being a $T$x11 matrix. Each column of textit$PC$ gives us the $T$ observations for each principal component. Since $(PC)'(PC) = T\Lambda$, then each principal component has variance equal to the associated eigenvalue, and it is uncorrelated with all the other principal components. Furthermore, the sum of the variances of the principal components is equal to the sum of variances of the original data set $X$ because both are equal to $trace(V)$. If we order the eigenvalues in $\Lambda$ from the largest to the smallest, and take the asociated eigenvectors to form the columns in $W$, the first principal component will be the one with the highest variance and thus the one with the highest explanatory power on the fluctuations exhibited by the variables in $X$ throughout the sample. The original data $X$ can be exactly represented in terms of the principal components by $X = (PC)W'$. If we take the subset of the first $r$ principal components (a $T$x$r$ matrix $PC^*$) and the first $r$ columns of $W$ (a $k$x$r$ matrix $W^*$), the representation,

$$X = (PC*)(W*)', \tag{5}$$

will hold as an approximation. We will use below such a representation for $r = 1$.

As shown in Table 7, the first principal component explains 65% of the fluctuations in the weekly changes of the 11 sectorial indices, a confirmation that there is strong commonality among the sectors. This is higher than the one estimated by Berndt and Obreja (2010) for European firms during the 2003 to 2008 period, but it is close to the average explanatory power estimated by Chen and Härdle (2015) for the pre- (58.7%) and post-crisis periods (72.3%).

**Table 7.** Principal component estimation from sectorial indices.

| Eigenvalue Order | Eigenvalue | Cumulative Variance | Sector | First Eigenv. | Second Eigenv. | Third Eigenv. | Fourth Eigenv. |
|---|---|---|---|---|---|---|---|
| 1 | 0.0133 | 0.654 | BM | 0.299 | −0.079 | 0.086 | 0.121 |
| 2 | 0.0014 | 0.722 | CG | 0.291 | −0.055 | 0.023 | 0.033 |
| 3 | 0.0011 | 0.776 | CS | 0.274 | −0.148 | −0.107 | −0.113 |
| 4 | 0.0010 | 0.827 | EN | 0.311 | −0.032 | 0.004 | −0.059 |
| 5 | 0.0008 | 0.866 | FIN | 0.352 | 0.123 | 0.097 | −0.086 |
| 6 | 0.0007 | 0.899 | GOV | 0.380 | 0.791 | 0.102 | −0.077 |
| 7 | 0.0005 | 0.926 | HC | 0.200 | −0.448 | 0.716 | −0.193 |
| 8 | 0.0005 | 0.951 | IND | 0.309 | −0.051 | 0.084 | 0.012 |
| 9 | 0.0004 | 0.968 | TEC | 0.234 | −0.123 | −0.055 | 0.915 |
| 10 | 0.0003 | 0.985 | TEL | 0.363 | −0.330 | −0.661 | −0.275 |
| 11 | 0.0003 | 1 | UTI | 0.251 | −0.001 | 0.023 | −0.068 |

Note: The table shows the eigenvalues of the matrix of standardized weekly changes in the sectorial credit indices, the cumulative variance explained by the eigenvectors, and the coefficients of each of the first four eigenvectors on the 11 sectorial indices. The order of the sectors is shown in column 4.

Cumulative percentage of total variance explained by the first two principal components is 72%, being 78% for the first three principal components, and 83% for the first four. To explain a percentage of variance of the order of 90% or 95%, we would need a relatively large number of components. The principal component loadings show that the first principal component is an approximate average of CDS returns over all the sectors, although with a slightly weaker presence of the health care and technology sectors (column 5 in Table 7). The second component has the larger loadings in the government sector, while the third component puts a heavier weight on health care

and telecommunications, and the fourth principal component is centered on technology (columns 7 and 8). Precisely those sectors with a lower representation in the first principal component, health care and technology, dominate the third and fourth principal components, respectively, while the second component is dominated by the government sector.[3] Principal components after the first four are much harder to interpret. Hence, the principal components with the higher explanatory power after the first one are essentially made up by sector-specific elements. Since any estimate of a global risk factor should avoid embedding idiosyncratic elements, and these are present in the successive principal components, we decide to stick to just the first principal component as the estimate of a latent global credit risk factor, capturing two thirds of the variance in the set of eleven sectorial credit indices.[4] Thus, if $CP_1$ denotes the principal component associated with the largest eigenvalue of the correlation matrix of weekly changes in the sectorial indices, we obtain the *GRF* estimate as

$$GRF_1 = 1.0, \tag{6}$$

$$GRF_t = GRF_{t-1} + CP_{1t}, t = 2, 3, ..., T, \tag{7}$$

where we have normalized the first observation, since principal components are identified except by a multiplicative constant. By construction, such a risk factor uses information on the set of CDS spreads trading at each point in time for all firms in the different sectors in all regions.

The evolution of the explanatory power of the principal components throughout the sample gives us an estimate of the way the degree of commonality has evolved over time. In so far as the effects of the financial and economic crisis were felt over the whole economy, we should expect to see the common factors increasing their relevance in that period of time, and dominating sector-specific risk elements. To check that hypothesis, we follow Eichengreen et al. (2012) to use annual windows to estimate the percentage of total variance in the weekly fluctuations set of 11 sectorial indices that is explained by the first principal component, as well as by the first two-, three-, and four-principal components (Figure 2).

At the beginning of 2007, the first common factor explained almost 40% of the total variation in sectorial CDS indices, with the first four factors explaining 75% of total variance. The explanatory power of the first principal component sharply jumped from 32% on 13 July 2007, to 62% on 20 August 2007 at the outbreak of the subprime crisis after the failure of three hedge funds at BNP Paribas. The increase in explanatory power did not stop there: the Bear Stearns rescue on March 2008 produced a sharp increase in the perception of risk across the economy, as reflected in the increase in commonality among all sectors of activity. Consequently, the explanatory power of the first common factor continued on a gradual increasing trend to a local maximum of 77% in the week of 9 May 2008. The maintained high commonality of risk after May 2008 could have been taken at that time, well in advance of the Lehman crisis, as an indication of potential future problems. These results are comparable to those obtained by Berndt and Obreja (2010) and Chen and Härdle (2015), among others.

---

[3] It is interesting to note that the government sector seems to have a strong specific behavior that explains its association with the second principal component in spite of having a loading in the first component in line with that of the other sectors.
[4] A decision in line with other authors, like Rodríguez-Moreno and Peña (2013).

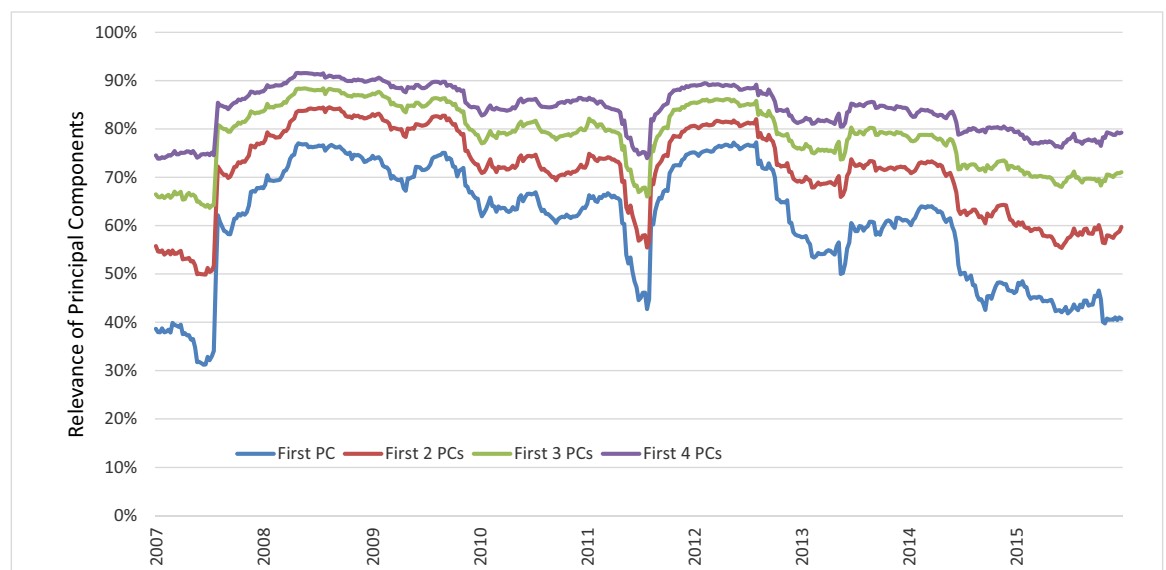

**Figure 2.** Cumulative information content in the first four principal components of sectorial returns. Weekly data: January 2007–December 2015. The figure shows the percent variance of the set of sectorial credit returns that is explained by the first k principal components of sectorial credit indices, $k = 1, 2, 3, 4$.

A sharp decrease was again observed in March 2011. On 11 March, the EU decided to allow the European Financial Stability Facility (EFSF) to buy debt in primary markets up to a 440-billion-euros ceiling. It also resolved to cut the rates and extend the maturities of the emergency loans to Greece. On the 21st, the EU summit agreed on a permanent bailout mechanism for the region to lend up to 50 billion euros starting in May 2013. The commonality in sectorial CDS indices declined from their peaks but remained at the post-Bear Stearns elevated levels, indicating that risk was widespread across the sectors. The explanatory power of the first common factor fell drastically to 43% on 5 August 2011. The bail out of Portugal on May 5 and the rating cut for Greece on 13 June did not have a visible effect on the explanatory power of the common factors. On the other hand, the downgrade of US debt on 5 August, the deterioration of the economic situation in the US, the alarm of a potentially catastrophic credit crisis in Europe and the downgrade of sovereign debt in southern European countries explain the increase in the commonality of risk observed in the second part of the year to levels of 75% by the end of 2011. From that point on, the explanatory power gradually decreased, staying at levels of 60% over 2013, to come down again to around 40% at the end of our sample.

*4.2. The Information Content of the Global Risk Factor*

Table 8 shows the regression estimates explaining sectorial credit indices with the Global Risk Factor as the single explanatory variable,

$$I_t = \beta_0 + \beta_1 GRF_t + u_t. \tag{8}$$

**Table 8.** One-factor Global Risk Factor (GRF) regressions explaining sectorial credit indices.

| Factor: GRF | Beta | adj R2 | $\sigma*100$ | Jarque–Bera | LBQ(1) | LBQ(4) | LBQ(12) | Arch Test | ADF(1) |
|---|---|---|---|---|---|---|---|---|---|
| BM | 0.30 | 0.70 | 2.28 | 30.1 | 0.0 | 3.2 | 19.9 | 9.4 | −16.6 |
| CG | 0.30 | 0.78 | 1.80 | 51.7 | 9.2 | 19.2 | 31.4 | 6.6 | −17.0 |
| CS | 0.28 | 0.66 | 2.29 | 132.7 | 0.8 | 5.0 | 32.6 | 26.4 | −16.0 |
| EN | 0.31 | 0.70 | 2.36 | 219.5 | 40.4 | 42.7 | 47.0 | 60.7 | −14.3 |
| FIN | 0.36 | 0.79 | 2.11 | 1682.1 | 72.0 | 81.6 | 95.0 | 182.1 | −12.9 |
| GOV | 0.38 | 0.62 | 3.32 | 991.4 | 7.9 | 10.5 | 19.7 | 2.8 | −15.8 |
| HC | 0.21 | 0.35 | 3.19 | 313.7 | 0.5 | 11.4 | 17.2 | 0.2 | −18.0 |
| IND | 0.31 | 0.79 | 1.84 | 52.6 | 0.4 | 12.6 | 23.3 | 3.5 | −17.8 |
| TEC | 0.24 | 0.44 | 3.06 | 42.7 | 0.7 | 6.3 | 22.2 | 40.4 | −16.9 |
| TEL | 0.36 | 0.65 | 3.04 | 136.4 | 1.2 | 6.7 | 26.2 | 16.1 | −16.7 |
| UTI | 0.26 | 0.71 | 1.86 | 115.5 | 9.5 | 12.4 | 38.1 | 25.6 | −15.6 |

Note: The table shows slope estimates, R2 statistics and the standard deviation of residuals in a least-squares regression of each sectorial index on GRF as the single explanatory variable. All regressions are estimated on weekly differences of both variables. The remaining columns show statistics for residual analysis. Jarque–Bera denotes the Normality statistic of that name, LBQ denotes the Ljung–Box Q statistic to test for residual autocorrelation, applied at at 1, 4 and 12 lags, Arch test denotes a test for first-order ARCH dependence, through the application of the LBQ test to the squared residuals, and ADF(1) is the Augmented Dickey–Fuller statistic including one lag of the dependent variable in the unit root regression. Critical values at 95% are: 5.86 for Jarque–Bera test; 3.84, 9.49 and 21.03 for Ljung–Box test at 1, 4 and 12 lags, respectively, 3.84 for the Arch test and −2.87 for the ADF test.

The fact that the global risk factor contains a good deal of information on fluctuations in sectorial CDS returns is to be expected, but it is surprising that it contains so much more explanatory power than credit market indices like iTraxx. The median R-squared is 0.70 for the global risk factor and 0.39 for the iTraxx index. Its high information content may arise because by averaging CDS spreads over the sectors, the first principal component incorporates some aspects of the credit market that might be sector specific and not incorporated in standard credit indices. Furthermore, the iTraxx may contain some idiosyncratic component unrelated to any specific sector, as reflected in the fact that it often presents deviations from the theoretical price that could be estimated from prices for its constituents, which could weaken its correlation with the sectorial credit indices. The high explanatory power on CDS issues from all sectors and geographical areas also justifies the interpretation of the first principal component as representing a global risk factor.

Incidentally, augmented Dickey–Fuller statistics for the residuals from regressions of sectorial indices either on iTraxx or on the global risk factor overwhelmingly reject the null hypothesis of a unit root in all sectors. That means that both sets of regressions can be interpreted as cointegrating regressions, with sectorial indices sharing the same stochastic trends as iTraxx or the global risk factor, differences between them being short-lived. It also means that the use of either iTraxx or the global risk factor to explain or predict sectorial indices should be done through an error correction model. However, the difference in R-squared statistics means that the global risk factor tracks sectorial indices much better than the iTraxx index.

It is also striking that our estimated global risk factor seems to incorporate the information contained in MSCI indices on the credit market, the MSCI indices not adding any information content to the global risk factor to explain the sectorial credit indices,

$$I_t = \beta_0 + \sum_{i=1}^{11} \beta_i MSCI_{it} + \gamma GRF_t + u_t, \tag{9}$$

as shown by R2 statistics in column 4 of Table 5. Even if we estimate a regression to explain sectorial credit indices using all the indicators described in Section 3, R-squared statistics do not reach the explanatory power attained by the global risk factor (column 5).[5]

Explaining sectorial credit indices with the GRF yield R-squared statistics up to twice as high as those obtained when we explain the sectorial indices with iTraxx, even though the time profiles of both indices is quite similar. Figure 3 shows that the weekly differences in both indices are less than perfectly correlated. Full-sample correlation is, in fact, 0.68. What disparate information is contained in the global risk factor? A first piece of evidence comes from the regression that adds the global risk factor to the MSCI indices to explain sectorial indices in column 8. We obtain in that model the same fit as with the global risk factor alone, showing that the latter embeds all the information provided by MSCI indices on sectorial risk credit. That is one important difference between the information contents of iTraxx and the global risk factor.

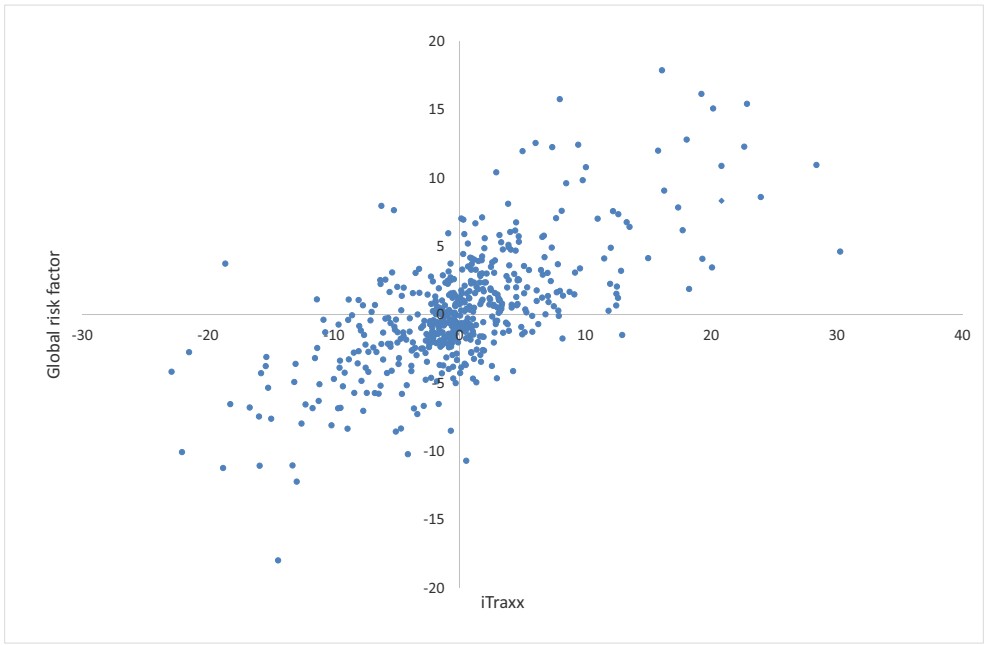

**Figure 3.** Weekly changes in global risk factor (GRF) and iTraxx. The scatterplot shows the weekly changes in the estimated global risk factor against those in iTraxx.

Further evidence on the information contained in GRF emerges from considering the residuals $\hat{u}_t$ from a regression of the GRF on iTraxx: $GRF_t = \beta_0 + \beta_1 iTraxx + u_t$. Such residuals give us the component $\widetilde{GRF}$ of our estimated global risk factor that is uncorrelated with iTraxx. Thus, if we now correlate $\widetilde{GRF}$ with the macroeconomic and financial indicators, we can have some idea of the type of information captured by GRF that is not contained in iTraxx, and Table 9 displays the correlations that are higher than 0.20 in absolute value. We find a negative association between this component of the global risk factor and three financial variables: the spread of the 3-month Libor rate over the USD Overnight Indexed Swap (OIS), the 1-year euro swap rate, and the 1-year US swap rate. Being a global index, it also presents a positive correlation with two credit indices, the Japanese iTraxx index, and the high yield CDX index. Finally, it has negative correlation with all MSCI indices, being above 0.20 with the materials, industrials, telecommunications and financial MSCI indices. Correlation signs are the same as those obtained for the global risk factor and these same indicators. This means that the global risk factor contains information on these indicators beyond the correlation shown by the iTraxx index.

---

[5]   In consistency with CDS spread data, we use the financial and macroeconomic indicators in weekly differences. As shown in Tables 1 and 2, weekly changes are stationary for all the indicators.

**Table 9.** Correlations with $\widetilde{GRF}$.

| Variable | Correlation | Variable | Correlation | Variable | Correlation |
|---|---|---|---|---|---|
| *Libor–OIS spread* | −0.327 | *iTraxx Japan* | −0.212 | *MSCI financials* | −0.232 |
| *Euro swap 1y* | −0.204 | *high yield CDX* | 0.225 | *MSCI Industrials* | −0.208 |
| *US swap 1y* | −0.255 | *MSCI basic materials* | −0.209 | *MSCI telecomm. services* | −0.204 |

Note: The table shows the correlations between the $\widetilde{GRF}$ component of the Global Risk Factor and the economic and financial indicators, which are higher than 0.20 in absolute value.

## 5. Systemic, Sectorial and Idiosyncratic Components of Credit Risk

We have proposed a simple method to estimate a global credit risk factor, GRF, using data on CDS spreads from a large sample of firms from different sectors and regions. Thus, our risk factor summarizes information across the whole credit market, and we have shown that it has a high explanatory power on the time behavior of sectorial indices, well above the information content in market indices like iTraxx and CDX. Furthermore, it seems to incorporate the stock market information in MSCI indices. Being such an efficient summary of the information on CDS spreads, it seems sensible to consider GRF as an indicator of systemic credit risk. We now proceed to evaluate the extent to which the credit risk of the different sectors is systematic or idiosyncratic in nature, a most relevant question for risk management. In the case of individual firms, we will decompose risk into systemic, sectorial and idiosyncratic components.

### 5.1. Sectorial Portfolios

The sectors with more important systemic risk will be those whose index moves closer to the global credit risk factor. Table 5 shows that the global risk factor has an R-squared above 70% over the whole period 2006–2015 for a number of sectorial indices. The appearance of the financial sector in this group is in line with the remarks made in Moody's (Munves 2008) and Basel (BCBS 2011) papers. In the last one, the Basel Committee proposed a specific increase in the estimated value of asset correlation for the financial sector when calculating the level of regulatory capital required.[6] The high R-squared for the industrial sector possibly reflects the impact of the global financial crisis in the real economy, since the industrial sector is distinctively dependent on capital to finance its their long-run investments, as well as reflecting the impact of the increased deterioration in the global housing market. The inclusion of consumer goods among the more systemic credit sectors can be due to the cyclical nature of the solvency of consumer credit.

Along the same line of reasoning, health care and technology would be the two least systemic sectors, in consistency with their lower correlations with other sectorial indices that we already saw in Table 4. The systemic nature of the health care sector is not surprising, taking into account the robust growth that it is experiencing around the world. This is especially the case in the developed countries (most of our sample) as the population of these countries is getting older, with more economic resources and a greater demand for health care services to achieve a better quality of life. As a consequence, the health care sector has been less influenced by the recent crisis. The specific nature of innovation in this industry, which has a life cycle very different from the other sectors of the economy, may explain the characterization of the technology sector as being less systemic than the rest.

---

[6] That correlation was set at 30%, up from the previous value of 24%, while the 24% correlation was kept for the rest of corporate sectors.

Since the fitted value yields the systemic component of risk, the residuals in the regressions of sectorial credit indices on the global risk factor can then be interpreted as the idiosyncratic component of credit risk in each sector. A perfectly adequate credit risk factor would capture all the commonality across sectors and hence the residuals should be sector-specific, with low correlations between them, in the spirit of idiosyncratic components. Indeed, the median absolute correlation between the residuals for any two sectors is just 0.094. Being small, it is less than fully satisfactory because it means that about half of the 55 correlations between sectorial idiosyncratic components are statistically significant, but the highest correlation is 0.29, and the 90% percentil is just 0.21. These correlations suggest that there might be some additional common element among the sectorial credit indices that is not captured by the global risk factor, although its explanatory power does not seem to be too large.

Figure 4 shows the explanatory power (R-squared statistic) attained by the global risk factor in 52-week rolling window regressions of the sectorial indices, which can be taken as a measure of the relevance of the systemic component of credit risk across sectors. They all show a similar pattern, except for the health care and technological sectors, which show a more disparate behavior. The relevance over time of the idiosyncratic component would be estimated as 1.0 minus the R-squared in Figure 4. We can see that, as a general rule, the systemic components become more relevant in stressed periods, when CDS spreads increase, while idiosyncratic components of risk become more important in calm periods. This must be reflecting increased correlations across sectorial indices in stressed periods and lower correlations in tranquil periods.

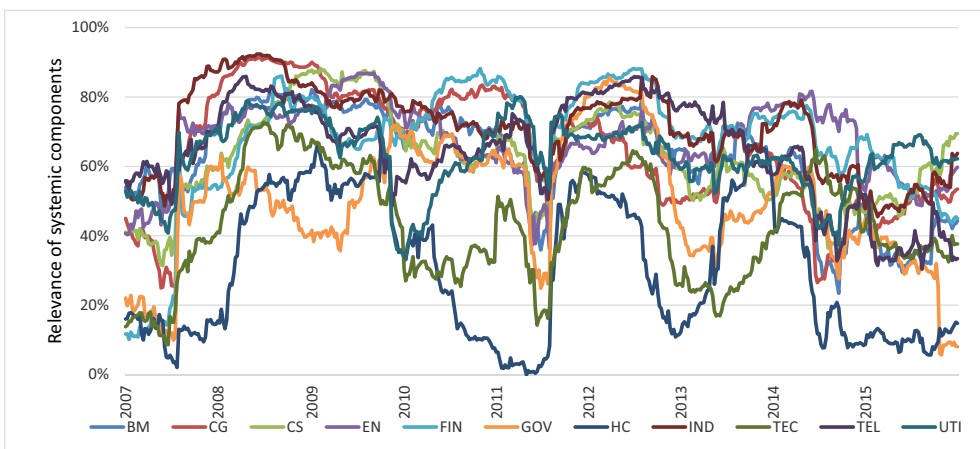

**Figure 4.** Relevance of systemic sectorial components. The figure shows the evolution over time of the relevance of the systemic component of risk in each sector, as a percentage of total credit risk. Weekly data: January 2007–December 2015. BM = Basic materials, CG = Consumer goods, CS = Consumer services, EN = Energy, FIN = Financials, GOV = Government, HC = Health care, IND = Industrials, TEC = Technology, TEL = Telecommunication services, and UTI = Utilities.

*5.2. Decomposition of Risk at the Level of the Firm*

In the previous section, we have analyzed the nature of credit risk in sectorial portfolios. Such information is needed for a rigorous asset allocation of credit among sectors. We now come down to the analysis of the characteristics of risk in some specific sectors, which should be the guide for asset allocation decisions inside a given sector. We want to measure to what extent firms in the sector are subject to systemic risk as well as to sectorial risk and what the relevance of idiosyncratic risk is. We count as systemic risk events that have influence across the global credit markets. By sectorial risk, we understand events that affect all firms in the sector, with no essential effect elsewhere. The idiosyncratic component of risk is obtained as what is left of each firm's CDS return after extracting the systemic and sectorial components of risk. Such evaluation of the relevance of risk components has obvious implications on the asset allocation strategy by a given financial institution that wants to diversify its credit portfolio in that sector. In designing their credit policy, financial institutions

should avoid firms with a large systemic risk component in favor of those with larger idiosyncratic risk components, always trying to form sufficiently diversified portfolios. As a byproduct, we also want to analyze whether the risk structure is common to a given sector in different geographical areas, such as the financial sectors of Europe and North America. That might suggest that sectorial characteristics are possibly more important than geographical characteristics in determining CDS spreads.

To estimate the systemic component of risk, we will use the same approach as with the sectorial indices: for each firm *i* in sector *j*, we estimate a regression of its CDS spread on the global risk factor:

$$CDS_{ijt} = \beta_0 + \beta_1 GRF_t + \mu_t, \ t = 1, 2 \ldots, T, \tag{10}$$

taking the fitted value as the systemic component in the CDS spread, and the R-squared as a measure of the relevance of the systemic component of risk for that firm. After that, we run a second regression on the global risk factor plus the first intra-sector principal component of CDS spreads,

$$CDS_{ijt} = \beta_0 + \beta_1 GRF_t + PC_{jt} + \mu_t, \ t = 1, 2 \ldots, T. \tag{11}$$

All of these are least-squares, single equation estimates. For each sector, this principal component will contain some features common across firms in the sector, possibly together with some elements of systemic risk. If we subtract the adjusted R-squared from both regressions, we will have an estimate of the relevance of sector-specific risk. The residual in the last regression can be taken as an estimate of the unobserved idiosyncratic risk component, and 1 minus the adjusted Rsquared in that regression is an estimate of the relevance of the idiosyncratic component of risk for that firm.

### 5.2.1. European Financial and Industrial Sectors

BCBS (2012) identifies five broad categories of factors that influence global systemic importance. The selected indicators reflect the size of banks, their interconnectedness, the lack of readily available substitutes or financial institution infrastructure for the services they provide, their global (cross-jurisdictional) activity and their complexity. Table 10 shows an example of the interconnectedness in the European financial sector with a decomposition of CDS risk for its firms. Systemic credit risk factors account for between 8% and 58% of total risk in the firms in this sector, and the intra-sector first principal component explains between 6% and 86% of CDS risk. Sectorial risk falls between 0% and 44% and idiosyncratic risk is between 13% and 92%. In terms of median values, systemic risk accounts for 42% of total risk, sectorial risk is 21% and idiosyncratic risk 30% of total risk. In half of the firms, 26 out of 52, the systemic component of risk is the most important, whereas in 22 firms, the idiosyncratic component of risk is highest.

These figures are similar to those shown in the European industrial sector (Table 11), although in the financial sector the idiosyncratic component is much more important. In 22 of the 26 European industrial firms in our sample, the systemic component of risk was the most important, with the idiosyncratic component being highest in just four firms.

**Table 10.** Credit risk decomposition for the European financial sector.

| Firm (1) | Systemic Risk (2) | Sectorial PC (3) | Both (4) | Sectorial Risk (5) | Idiosyncratic Risk (6) |
|---|---|---|---|---|---|
| AXA | **58%** | 75% | 76% | 19% | 24% |
| Legal & Gen Gp PLC | **57%** | 69% | 71% | 15% | 29% |
| Mediobanca SpA | **56%** | 80% | 80% | 25% | 20% |
| Prudential PLC | **55%** | 66% | 69% | 14% | 31% |
| Assicurazioni Generali S p A | **52%** | 81% | 81% | 29% | 19% |
| Aviva plc | **52%** | 75% | 75% | 23% | 25% |
| Rabobank Nederland | **50%** | 80% | 80% | 30% | 20% |
| ING Bk N V | **50%** | 86% | 86% | 36% | 14% |
| Old Mut plc | **50%** | 45% | 54% | 3% | 47% |
| HSBC Bk plc | **50%** | 81% | 81% | 31% | 19% |
| Royal & Sun Alliance Ins PLC | **49%** | 70% | 70% | 21% | 30% |
| Bca Pop di Milano Soc Coop a r l | **49%** | 76% | 76% | **27%** | 24% |
| Munich Re | **49%** | 78% | 78% | 29% | 22% |
| Bca Monte dei Paschi di Siena S p A | **48%** | 76% | 76% | 29% | 24% |
| Bca Naz del Lavoro S p A | **47%** | 80% | 80% | 33% | 20% |
| Std Chartered Bk | **46%** | 73% | 73% | 27% | 27% |
| Societe Generale | **45%** | 83% | 84% | 39% | 16% |
| STANDARD CHARTERED PLC | **45%** | 71% | 71% | 26% | 29% |
| BNP Paribas | **45%** | 86% | 87% | 41% | 13% |
| ACE Ltd | 44% | 35% | 45% | 1% | **55%** |
| Aegon N.V. | **44%** | 60% | 60% | 17% | 40% |
| Skandinaviska Enskilda Banken AB | **43%** | 60% | 60% | 17% | 40% |
| UBS AG | **43%** | 78% | 78% | 35% | 22% |
| Cr Agricole SA | 43% | 84% | 86% | **43%** | 14% |
| Hammerson PLC | 42% | 29% | 42% | 0% | **58%** |
| Deutsche Bk AG | **42%** | 79% | 80% | 38% | 20% |
| Bco Bilbao Vizcaya Argentaria S A | **42%** | 76% | 77% | 35% | 23% |
| Cr LYONNAIS | **42%** | 82% | 84% | 42% | 16% |
| Raiffeisen Zentralbank Oesterreich AG | 42% | 46% | 49% | 7% | **51%** |
| Inv AB | 41% | 23% | 41% | 0% | **59%** |
| Commerzbank AG | 40% | 80% | 81% | **41%** | 19% |
| Danske Bk A S | 39% | 59% | 59% | 20% | **41%** |
| Nordea Bk AB | 38% | 57% | 57% | 18% | **43%** |
| KBC Bk | 38% | 57% | 57% | 19% | **43%** |
| Barclays Bk plc | 38% | 80% | 82% | **44%** | 18% |
| Bco Comercial Portugues SA | **36%** | 69% | 70% | 35% | 30% |
| Royal Bk of Scotland Pub Ltd Co | 36% | 75% | 77% | **41%** | 23% |
| Klepierre | 35% | 21% | 35% | 0% | **65%** |
| Gecina | 35% | 37% | 40% | 5% | **60%** |
| 3i Gp plc | 34% | 21% | 34% | 0% | **66%** |
| ISS Glob A S | 32% | 28% | 33% | 1% | **67%** |
| Bco de Sabadell S A | 31% | 51% | 51% | 20% | **49%** |
| Svenska Handelsbanken AB | 31% | 50% | 50% | 19% | **50%** |
| Dexia Cr Loc | 28% | 58% | 60% | 32% | **40%** |
| Bay Landbk Giroz | 26% | 50% | 51% | 24% | **50%** |
| Landbk Baden Wuertbg | 25% | 39% | 38% | 14% | **62%** |
| Nationwide Bldg Soc | 24% | 45% | 45% | 21% | **55%** |
| Brit Ld Co plc | 21% | 11% | 21% | 0% | **79%** |
| DZ Bk AG | 21% | 23% | 25% | 4% | **75%** |
| IKB Deutsche Industriebank AG | 14% | 22% | 22% | 8% | **78%** |
| Ld Secs PLC | 12% | 8% | 12% | 0% | **88%** |
| Storebrand ASA | 8% | 6% | 8% | 0% | **92%** |

Note: Column 1 shows the company name from Markit database. Column 2 displays the adjusted R-squared from a regression on the Global risk factor, which we take as a measure of the relevance of the systemic component. Column 3 shows the R-squared from a regression on the sectorial index. Column 4 shows the R-squared from a regression on both indices. The relevance of the sectorial component of risk is obatined as the difference between columns 4 and 2. The relevance of the idiosyncratic component of risk is obtained as 1 minus the R-squared in column 4. Bold figures indicate the most important factor the risk decomposition for each firm.

**Table 11.** Credit risk decomposition for the European industrial sector.

| Firm (1) | Systemic Risk (2) | Sectorial PC (3) | Both (4) | Sectorial Risk (5) | Idiosyncratic Risk (6) |
|---|---|---|---|---|---|
| Cie de St Gobain | **65%** | 79% | 79% | 13% | 21% |
| THALES | **62%** | 77% | 77% | 15% | 23% |
| Lafarge | **62%** | 74% | 74% | 13% | 26% |
| Vinci | **59%** | 74% | 74% | 15% | 26% |
| Adecco S A | **59%** | 69% | 69% | 10% | 31% |
| AB Volvo | **59%** | 74% | 74% | 15% | 26% |
| BAE Sys PLC | **56%** | 70% | 70% | 14% | 30% |
| ASSA ABLOY AB | **56%** | 59% | 60% | 4% | 40% |
| Atlas Copco AB | **56%** | 56% | 59% | 3% | 41% |
| Rexam plc | **55%** | 63% | 63% | 8% | 37% |
| Volvo Treas AB | **55%** | 66% | 66% | 12% | 34% |
| SCANIA AB | **55%** | 63% | 63% | 9% | 37% |
| Metso Corp | **54%** | 59% | 60% | 6% | 40% |
| Siemens AG | **53%** | 58% | 59% | 6% | 41% |
| Finmeccanica S p A | **52%** | 66% | 66% | 14% | 34% |
| ROLLSROYCE PLC | **51%** | 68% | 68% | 17% | 32% |
| Deutsche Lufthansa AG | **49%** | 63% | 64% | 15% | 36% |
| SOCIETE AIR FRANCE | **49%** | 60% | 60% | 12% | 40% |
| HeidelbergCement AG | **48%** | 60% | 60% | 12% | 40% |
| Securitas AB | **47%** | 56% | 56% | 9% | 44% |
| Deutsche Post AG | **46%** | 57% | 57% | 11% | 43% |
| ALSTOM | **44%** | 56% | 56% | 12% | 44% |
| Brit Awys plc | 44% | 52% | 52% | 8% | **48%** |
| AB SKF | 43% | 43% | 45% | 2% | **55%** |
| Smiths Gp Plc | 39% | 48% | 48% | 9% | **52%** |
| Rentokil Initial 1927 Plc | 25% | 31% | 31% | 6% | **69%** |

Note: Column 1 shows the company name from Markit database. Column 2 displays the adjusted R-squared from a regression on the Global risk factor, which we take as a measure of the relevance of the systemic component. Column 3 shows the R-squared from a regression on the sectorial index. Column 4 shows the R-squared from a regression on both indices. The relevance of the sectorial component of risk is obtained as the difference between columns 4 and 2. The relevance of the idiosyncratic component of risk is obtained as 1 minus the R-squared in column 4. Bold figures indicate the most important factor the risk decomposition for each firm.

### 5.2.2. North American Financial and Industrial Sectors

In terms of median R-squared values across North American financial firms, the systemic factor accounts for 38% of total CDS return risk, sectorial factors explain 12%, and firm-specific factors explain the largest amount, 48% of total CDS risk (Table 12). Thus, 50% of the credit risk in these firms has a systemic or sectorial nature, the other 50% being idiosyncratic, which leaves significant possibilities for portfolio diversification. These figures are very similar to those we obtained for the North American industrial sector.

For the North American industrial firms (Table 13), the systemic component explains 38% of credit risk, with the sectorial component accounting for 12% of risk and the idiosyncratic component explaining 46%, in median terms. In 21 out of the 37 firms, firm-specific factors are the most important component of risk, with systemic risk being the most important one in the remaining 16 firms.

This result once more suggests the difficulty in finding a successful hedge for undiversified positions in CDS from these firms, which might possibly be explained by the lack of liquidity of the CDS market.

**Table 12.** Credit risk decomposition for the North American financial sector.

| Firm (1) | Systemic Risk (2) | Sectorial PC (3) | Both (4) | Sectorial Risk (5) | Idiosyncratic Risk (6) |
|---|---|---|---|---|---|
| Simon Ppty. Gp. L P | **59%** | 64% | 67% | 8% | 33% |
| Simon Ppty. Gp. Inc. | **57%** | 60% | 63% | 6% | 37% |
| American Express Co. | **54%** | 77% | 77% | 23% | 23% |
| HARTFORD FINL. SERVICES GROUP INC | **54%** | 75% | 75% | 22% | 25% |
| Prudential Finl. Inc. | **53%** | 74% | 73% | 20% | 27% |
| MetLife Inc. | **53%** | 77% | 77% | 24% | 23% |
| Caterpillar Finl. Svcs Corp. | **53%** | 53% | 58% | 4% | 42% |
| ERP Oper. Ltd. Pship. | **53%** | 58% | 60% | 7% | 40% |
| Avalon Bay Cmntys Inc. | **53%** | 54% | 57% | 5% | 43% |
| Berkshire Hathaway Inc. | **52%** | 63% | 64% | 12% | 36% |
| General Electric Cap Corp. | **52%** | 69% | 69% | 18% | 31% |
| Lincoln Natl. Corp. | **50%** | 67% | 67% | 16% | 33% |
| John Deere Cap Corp. | **49%** | 51% | 54% | 5% | 46% |
| CNA Finl. Corp. | **48%** | 61% | 61% | 13% | 39% |
| Allstate Corp. | **48%** | 64% | 64% | 17% | 36% |
| HSBC Fin. Corp. | **47%** | 65% | 64% | 17% | 36% |
| INTL. LEASE FIN. CORP. | **45%** | 63% | 63% | 18% | 37% |
| DUKE Rlty. Ltd. PARTNERSHIP | 45% | 36% | 45% | 0% | **55%** |
| Mack Cali Rlty. LP | 44% | 39% | 45% | 1% | **55%** |
| CHUBB CORP. | **44%** | 60% | 60% | 17% | 40% |
| Boeing Cap Corp. | 44% | 47% | 49% | 6% | **51%** |
| JPMorgan Chase & Co. | **42%** | 64% | 64% | 22% | 36% |
| Cap One Finl. Corp. | **41%** | 64% | 65% | 24% | 35% |
| Goldman Sachs Gp. Inc. | **40%** | 63% | 64% | 24% | 36% |
| Liberty Mut. Ins. Co. | 39% | 55% | 55% | 15% | **45%** |
| Loews Corp. | 39% | 47% | 47% | 9% | **53%** |
| G A T X Corp. | 38% | 35% | 40% | 2% | **60%** |
| Bank of America Corp. | **38%** | 63% | 64% | 26% | 37% |
| Aon Corp. | 37% | 48% | 48% | 11% | **52%** |
| Natl. Rural Utils Coop. Fin. Corp. | 37% | 50% | 50% | 14% | **50%** |
| Citigroup Inc. | **36%** | 63% | 65% | 29% | 35% |
| Wells Fargo & Co. | **35%** | 64% | 66% | 31% | 34% |
| Morgan Stanley | 35% | 62% | 64% | 29% | **36%** |
| SEARS ROEBUCK Accep. CORP. | 32% | 31% | 34% | 2% | **66%** |
| American Express Cr. Corp. | 31% | 42% | 42% | 11% | **58%** |
| American Intl. Gp. Inc. | 31% | 49% | 49% | 19% | **51%** |
| Marsh & Mclennan Cos Inc. | 27% | 36% | 36% | 9% | **64%** |
| Toyota Mtr. Cr. Corp. | 27% | 18% | 26% | 0% | **74%** |
| EOP Oper. Ltd. Pship. | 27% | 27% | 29% | 2% | **71%** |
| MGIC Invt. Corp. | 26% | 43% | 44% | 18% | **56%** |
| Radian Asset Assurn. Inc. | 25% | 42% | 43% | 18% | **57%** |
| Radian Gp. Inc. | 25% | 43% | 44% | 19% | **56%** |
| HEALTHCARE Rlty. Tr. Inc. | 24% | 21% | 24% | 0% | **76%** |
| Safeco Corp. | 24% | 26% | 27% | 3% | **73%** |
| BROOKFIELD ASSET Mgmt. INC. | 18% | 8% | 21% | 2% | **79%** |
| Fairfax Finl. Hldgs. Ltd. | 18% | 19% | 19% | 2% | **81%** |
| American Finl. Gp. Inc. | 17% | 12% | 16% | 0% | **84%** |
| Odyssey Re Hldgs. Corp. | 15% | 20% | 20% | 4% | **80%** |
| MBIA Ins. Corp. | 15% | 31% | 33% | 18% | **67%** |
| MBIA Inc. | 13% | 32% | 36% | 23% | **64%** |
| Highwoods Rlty. LP | 7% | 6% | 7% | 0% | **93%** |
| Legg Mason Inc. | 2% | 1% | 3% | 1% | **97%** |

Note: Column 1 shows the company name from Markit database. Column 2 displays the adjusted R-squared from a regression on the Global risk factor, which we take as a measure of the relevance of the systemic component. Column 3 shows the R-squared from a regression on the sectorial index. Column 4 shows the R-squared from a regression on both indices. The relevance of the sectorial component of risk is obtained as the difference between columns 4 and 2. The relevance of the idiosyncratic component of risk is obtained as 1 minus the R-squared in column 4. Bold figures indicate the most important factor the risk decomposition for each firm.

**Table 13.** Credit risk decomposition for the North American industrial sector.

| Firm (1) | Systemic Risk (2) | Sectorial PC (3) | Both (4) | Sectorial Risk (5) | Idiosyncratic Risk (6) |
|---|---|---|---|---|---|
| Utd Tech. Corp. | **53%** | 70% | 70% | 17% | 30% |
| Caterpillar Inc. | **50%** | 71% | 71% | 21% | 29% |
| Deere & Co. | **48%** | 70% | 70% | 22% | 30% |
| Eaton Corp. | **48%** | 52% | 54% | 6% | 46% |
| Textron Finl. Corp. | **47%** | 53% | 54% | 7% | 46% |
| Gen Dynamics Corp. | **47%** | 71% | 71% | 25% | 29% |
| Cummins Inc. | **46%** | 54% | 55% | 9% | 45% |
| TEXTRON INC. | **46%** | 58% | 58% | 12% | 42% |
| Boeing Co. | **45%** | 66% | 66% | 21% | 34% |
| Arrow Electrs Inc. | **45%** | 58% | 58% | 14% | 42% |
| Emerson Elec. Co. | **44%** | 57% | 57% | 14% | 43% |
| Packaging Corp. Amer. | 44% | 51% | 52% | 8% | **48%** |
| Ryder Sys. Inc. | **43%** | 57% | 57% | 14% | 43% |
| Danaher Corp. | 42% | 54% | 54% | 12% | **46%** |
| Raytheon Co. | **40%** | 68% | 70% | 29% | 30% |
| Southwest Airls. Co. | 40% | 58% | 58% | 18% | **42%** |
| Lockheed Martin Corp. | **39%** | 65% | 66% | 27% | 34% |
| Owens IL Inc. | 38% | 43% | 44% | 6% | **56%** |
| Norfolk Sthn. Corp. | **38%** | 66% | 68% | 30% | 32% |
| Navistar Intl. Corp. | 37% | 34% | 39% | 1% | **61%** |
| Utd Rents Inc. | 37% | 41% | 42% | 5% | **58%** |
| CSX Corp. | **37%** | 63% | 64% | 28% | 36% |
| Sealed Air Corp. US | 36% | 49% | 49% | 13% | **51%** |
| FedEx Corp. | 36% | 58% | 59% | 23% | **41%** |
| Cdn Natl. Rwy Co. | 35% | 47% | 47% | 12% | **53%** |
| L 3 Comms Corp. | 33% | 43% | 43% | 10% | **57%** |
| R R Donnelley & Sons Co. | 28% | 40% | 40% | 12% | **60%** |
| 1st Data Corp. | 26% | 40% | 41% | 15% | **59%** |
| Waste Mgmt Inc. | 24% | 27% | 28% | 4% | **72%** |
| Rd King Infstruc. | 20% | 12% | 21% | 1% | **79%** |
| Owens Brockway Glass Container Inc. | 19% | 21% | 22% | 2% | **78%** |
| Rep Svcs Inc. | 19% | 22% | 23% | 3% | **77%** |
| JetBlue Awys Corp. | 11% | 14% | 14% | 2% | **86%** |
| Cooper Inds. Ltd. | 7% | 7% | 7% | 1% | **93%** |
| Sonoco Prods. Co. | 6% | 4% | 6% | 0% | **94%** |
| PHH Corp. | 3% | 2% | 2% | 0% | **98%** |
| Briggs & Stratton Corp. | 2% | 1% | 2% | 0% | **98%** |

Note: Column 1 shows the company name from Markit database. Column 2 displays the adjusted R-squared from a regression on the Global risk factor, which we take as a measure of the relevance of the systemic component. Column 3 shows the R-squared from a regression on the sectorial index. Column 4 shows the R-squared from a regression on both indices. The relevance of the sectorial component of risk is obtained as the difference between columns 4 and 2. The relevance of the idiosyncratic component of risk is obtained as 1 minus the R-squared in column 4. Bold figures indicate the most important factor the risk decomposition for each firm.

## 6. An Evaluation of the Proposed Methodology for Risk Decomposition

### 6.1. Two Validation Tests

The estimated idiosyncratic component of CDS risk turns out to be quite large in many firms, especially in the US market, so we should worry about the possibility that this might be due to measurement error in the idiosyncratic component, which could still contain some elements of systemic risk. To check on the effectiveness of our methodology to identify the idiosyncratic component of credit risk, we discuss now two related issues.

The first issue is whether our estimated sectorial component of risk is free from idiosyncratic features. CDS spreads for North American financial firms have a median pairwise linear correlation

coefficient of 0.49, while the median correlation between each firm in that sector and the financial sector credit index we constructed in Section 3.2 is 0.66, with a highest correlation of 0.78. Thus, North American financial firms share important elements of risk and they bear a relatively close association with the sectorial credit index, showing that there exists a well-defined sectorial component of risk.

The intrasector first principal component for these firms has a still higher correlation, of 0.88, with the credit index for the financial sector. Such correlation is unexpectedly high. The financial sector index of Section 3.2 is made up by the median spread negotiated each day in CDS by all firms in the financial sector from all the different geographical regions. Therefore, each daily observation on the financial sector index may come from a different financial firm, and even from a different country. On the other hand, the principal component for the North American financial sector is a linear combination of spreads from all CDSs traded each day by North American firms. Hence, it is some sort of average of these specific CDSs, all of them from the same geographical area. The two measures are different enough so that such a high correlation between them is remarkable. It shows that the average of CDS spreads that is embedded into the intra-sector principal component is quite successful at filtering out idiosyncratic components, essentially capturing the same sectorial features as the credit index for the global financial sector.

Strikingly enough, the European financial sector shares these characteristics: the correlation between our estimate of the sectorial component of risk and the financial sector credit index from Section 3.2 is 0.79.[7] Individual CDS spreads have moderately high correlations between them, with a median value of 0.67, and a median correlation of 0.72 with the financial sector credit index, with a maximum of 0.82. Again, the estimated sectorial component of risk is much closer to the sectorial credit index than CDS spreads for individual firms, showing that the former is quite free from idiosyncratic features.

The bottom line of this analysis is that we can indeed use the principal component methodology with data from a given geographical region to extract a sectorial component of risk that turns out to be similar to the sectorial credit index that can be obtained from all CDS trading in all regions. Our construction of the global sector factor is not directly responsible for this result. In fact, choosing the median of all CDS spreads traded each day over the world does not seem to be the most direct way to generate a high correlation with an average of sector spreads in a specific region. The implications are important. They suggest that the first intrasector principal component across firms is essentially free of firm idiosyncratic characteristics, thereby justifying our estimates of sectorial components of risk.

The second issue relates to whether our estimates of the idiosyncratic components of risk have the appropriate features. First of all, our estimates of the idiosyncratic components of risk turn out to be essentially uncorrelated across firms, which is a necessary condition for the interpretation we give to this component. There are 26 firms in the European industrial sector, 42 in the North American industrial sector, 52 in the European financial sector, and 52 firms in the North American financial sector. That amounts to 325 and 861 correlations between pairs of idiosyncratic components in the European and North American industrial sectors and 1326 correlations in the European and North American financial sectors. Median correlations are very low: −0.06, −0.02, −0.03 and −0.03, respectively. Ninety percent of them are below 0.26, 0.27, 0.24 and 0.18, respectively, in absolute value. These are all low levels that justify an interpretation of our estimated idiosyncratic components as being firm-specific in nature.

*6.2. Diversification Strategies Based on Estimated Idiosyncratic Components*

A further check on the nature of our estimated idiosyncratic components consists of examining possible diversification strategies. Portfolios made up of firms with a high idiosyncratic component should be hard to hedge unless we include a large number of such firms. For a given size, portfolios

---

[7]　Incidentally, remember that the financial sector credit index is the same for European and North American firms.

made up of firms with a low idiosyncratic component of risk should be much easier to hedge than portfolios of the same size made up of firms with low idiosyncratic components. Table 14 shows the reduction in variance of portfolios of different sizes when we hedge them taking a contrary position in Traxx. Portfolios in the table include groups of 5, 10 or 20 firms having either the highest or the lowest estimated idiosyncratic components. The results are as expected: the efficiency of the hedge, as measured by the reduction in variance, is higher for the portfolios made up of less idiosyncratic firms than for those built with the more idiosyncratic firms. In addition, the efficiency increases with the size of the portfolio, converging, as the number of firms grows, to the efficiency achieved when hedging the equally weighted portfolio for each sector.

**Table 14.** Variance reduction for sectorial portfolios.

| Sector | Number of Firms | More Idiosyncratic | Less Idiosyncratic | Equally Weighted Portfolio |
|---|---|---|---|---|
| **European industrial** | 5 firms | 45% | 63% | |
| | 10 firms | 57% | 64% | 64% |
| | 20 firms | 65% | 65% | |
| **North American industrial** | 5 firms | 8% | 46% | |
| | 10 firms | 23% | 47% | 52% |
| | 20 firms | 51% | 44% | |
| **European financial** | 5 firms | 23% | 55% | |
| | 10 firms | 40% | 58% | 60% |
| | 20 firms | 50% | 59% | |
| **North American financial** | 5 firms | 13% | 48% | |
| | 10 firms | 25% | 52% | 54% |
| | 20 firms | 40% | 54% | |

Note: The table shows the reduction in variance from alternative portfolios, when the hedge is constructed by taking a contrary position in the iTraxx index. Results are provided for portfolios of the 5, 10 and 30 firms with the highest or the lowest idiosyncratic components of risk in the four sectors considered, as well as for the equally weighted portfolio made up with all the firms in the sector.

The low correlation among the idiosyncratic components of individual firms and the good possibilities for hedging risk of a well-diversified sectorial portfolio suggest that our estimates of the idiosyncratic components of risk are appropriate.

*6.3. Idiosyncratic Risk and Lack of Liquidity*

What is behind the large idiosyncratic component of risk? A possible conjecture for the large size of idiosyncratic components of risk might be again that they are just a reflection of the low liquidity in some issues. To check on this assumption, we could try to relate the size of the estimated idiosyncratic risk with either the number of contributors giving prices to the 5-year CDS (Composite depth 5-yr.), the quality rating of the data provided by Markit, or the volatility of CDS returns. In the latter case, the argument would be that illiquid CDSs would often repeat prices in the Markit quotes, with the time series of CDS spreads then having a relatively low variance. Hence, we would expect a negative correlation between the size of the idiosyncratic component of risk and the volatility of CDS spreads. The correlation between the size of the idiosyncratic risk component and the annual volatility of CDS returns among European financial firms is equal to $-0.60$, being equal to $-0.46$ for North American financial firms. Thus, there seems to be, in fact, some evidence on the fact that the large size of the idiosyncratic risk component for some firms is in part due to the low liquidity of their CDSs.

**7. Robustness Tests**

To complement the evaluation of our methodology to estimate a Global Risk Factor, we consider now some alternative estimation approaches. As explained above, our main proposal consists of using as GRF the first principal component of the weekly changes in the 11 sectorial indices. These

indices were previously constructed by calculating the median CDS spread traded each day in that sector across firms in all regions, and then taking the weekly average of the daily observations for the sectorial index. Working in first differences avoids the nonstationarity of CDS spreads, but it is well known that the principal component methodology can give good results when working with nonstationary data if they maintain common sources of stochastic trends. Thus, we computed the median CDS spread traded each day for all firms in a given sector, and then the weekly average of those observations. Figure 5 shows this estimate, GRFlevels, jointly with the original GRF estimated from the weekly changes in sectorial indices. The levels of the two estimates are not the same, but the time pattern is very similar.

A different approach would consider to exclude the Financial and Government sector to estimate a Corporate GRF using the same methodology described in Section 4. The result is also shown in Figure 5, and displays again a time pattern similar to the other two estimates.

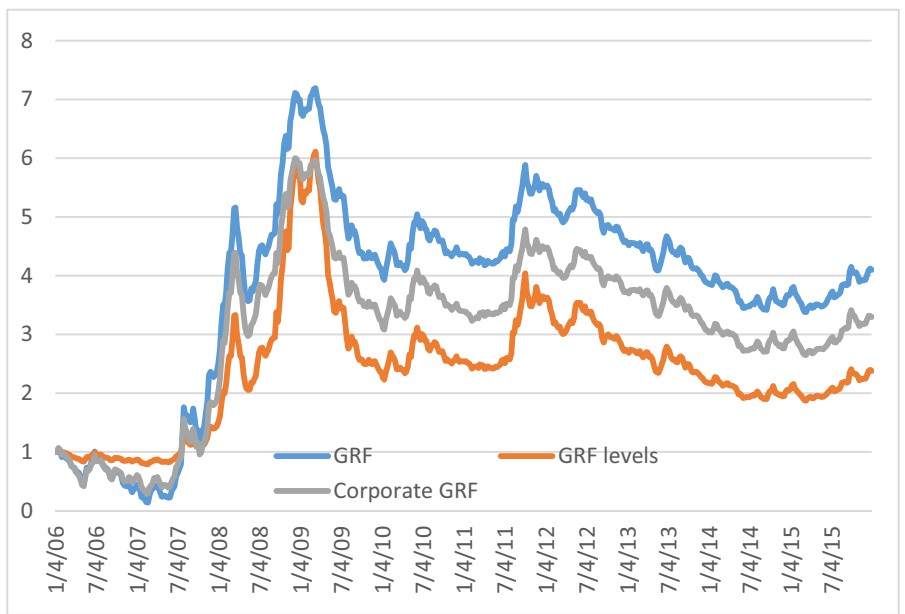

**Figure 5.** Global Risk Factor (GRF): alternative estimates. Note: the figure shows three alternative estimates of the global risk factor: Our main estimation proposal (GRF), the estimate obtained from CDS spreads in levels (GRF levels), and the Corporate GRF (Corporate GRF).

Table 15 shows linear correlation coefficients between weekly changes of these three estimates of a Global Risk Factor. We can see that, in spite of the differences in levels observed in Figure 5, the information content of the three estimates is similar. The same conclusion arises from Table 16, which shows $R^2$ statistics from least-squares regressions of the sectorial indices on each of the three estimates. If anything, the GRFlevels estimate could be said to have a slightly lower explanatory power across sectors. We should also examine the implications of using these alternative estimates for the risk decomposition we have presented in previous sections. Table 17 shows that, for the industrial and financial sectors of US and Europe, estimates of the relevance of the risk components obtained with the original and the Corporate GRF estimates would be very similar.

**Table 15.** Correlations between alternative GRF estimates.

| Risk Indicator | GRF | GRFlevels | Corp GRF |
|---|---|---|---|
| GRF | 1 | 0.986 | 0.908 |
| GRFlevels | | 1 | 0.913 |
| Corp GRF | | | 1 |

Note: The table shows correlations between weekly changes in the alternative estimates of the Global Risk Factor.

**Table 16.** Explanatory power of alternative GRF estimates.

| Sector | GRF | GRFlevels | Corp GRF |
|--------|-----|-----------|----------|
| BM | 0.70 | 0.67 | 0.70 |
| CG | 0.75 | 0.73 | 0.76 |
| CS | 0.66 | 0.64 | 0.66 |
| EN | 0.63 | 0.60 | 0.62 |
| FIN | 0.71 | 0.68 | 0.70 |
| GOV | 0.64 | 0.63 | 0.64 |
| HC | 0.37 | 0.35 | 0.38 |
| IND | 0.80 | 0.75 | 0.79 |
| TEC | 0.46 | 0.45 | 0.47 |
| TEL | 0.67 | 0.66 | 0.67 |
| UTI | 0.67 | 0.63 | 0.66 |
| Median | 0.67 | 0.64 | 0.66 |

Note: The table shows R2 statistics from regressions of the sectorial credit indices on alternative estimates of the Global Risk Factor. The regressions are estimated using weekly changes of all variables. BM = Basic materials, CG = Consumer goods, CS = Consumer services, EN = Energy, FIN = Financials, GOV = Government, HC = Health care, IND = Industrials, TEC = Technology, TEL = Telecommunication services, and UTI = Utilities.

**Table 17.** Correlation coefficients between alternative estimates of the relevance of risk components.

| Sector | Systemic | Idiosyncratic |
|--------|----------|---------------|
| US Industrial | 0.997 | 0.942 |
| Europe Industrial | 0.995 | 0.993 |
| US Financial | 0.996 | 0.981 |
| Europe Financial | 0.988 | 0.982 |

Note: The table shows linear correlation coefficients between the estimates of the relevance of the systemic and idiosyncratic components of risk obtained from using the original GRF or the Corporate GRF to obtain such estimates.

## 8. Synthetic Factor Regressions

To further clarify the relative importance of the different possible sources of risk in determining the different risk components for individual firms, we will use a synthetic factor for each group of indicators in Section 3: MSCI sectorial indices as indicators from equity markets, and macroeconomic, risk aversion, and financial indicators, all of them in weekly differences to avoid non-stationarity. We are going to consider in this section all the firms in our sample from the industrial and financial sectors of the US and Europe, and we will examine the relationship between the relevance of each tipe of risk and the sensitivity to each of the four types of risk: equity, macroeconomic, risk aversion and financial.

Some indicators display significant correlations, as can be seen in Table 18. Weekly changes between swap rates at different maturities or across countries are highly correlated, and these rates are also correlated with those for government debt. Short term rates and term structure slopes of US and Europe are also highly correlated, and the same is observed for stock market or credit market volatilities of US and Europe. Correlations between weekly changes in the ten sectorial MSIC indices fall between 0.54 and 0.93, with a median correlation of 0.75 (not shown in the table).

To avoid collinearity and to exploit optimally this common information, we obtain synthetic indicators by taking the first principal component in each group of indicators. The synthetic equity indicator has a correlation above 0.75 with all sectorial MSCI indices except healthcare, so that it captures the general evolution of stock prices. The risk synthetic indicator is an approximate average of all the implied volatility variables, with a much smaller weight of the liquidity indicators. Interestingly enough, the risk indicator shows the highest correlation with swaption implied volatilities, both in euros and US dollars, and stock market volatilities, well above its correlation with credit or exchange rate volatility. The macro synthetic indicator essentially captures a positive dependence on Government bond yields from US and Europe as well as with term structure slopes. Correlations of

this synthetic indicator with exchange rates and term structure curvatures are much lower. Finally, the synthetic financial indicator essentially captures risk in medium- and long-term swap rates in the US and Europe.

**Table 18.** Correlation among financial risk indicators.

| Panel a | | | | | |
|---|---|---|---|---|---|
| **US** | **1y Swap** | **5y Swap** | **10y Swap** | **5y Bond** | **10y Bond** |
| 1y swap | 1 | 0.66 | 0.49 | 0.58 | 0.44 |
| 5y swap | 0.66 | 1 | 0.94 | 0.95 | 0.89 |
| 10y swap | | | | 0.90 | 0.95 |
| 5y bond | | | | | 0.90 |
| Europe | 1y swap | 5y swap | 10y swap | 5y bond | 10y bond |
| 1y swap | 1 | 0.72 | 0.51 | 0.63 | 0.45 |
| 5y swap | | 1 | 0.91 | 0.91 | 0.83 |
| 10y swap | | | 1 | 0.85 | 0.91 |
| 5y bond | | | | 1 | 0.89 |
| 10y bond | | | | | 1 |
| Japan | 1y swap | 5y swap | 10y swap | | |
| 1y swap | 1 | 0.69 | 0.46 | | |
| 5y swap | 0.69 | 1 | 0.89 | | |
| 10y swap | 0.44 | 0.83 | 0.90 | | |

| Panel b | | | | | |
|---|---|---|---|---|---|
| **Variable** | **(US, Europe)** | **(US, Japan)** | **(Europe, Japan)** | **Variables** | **(US, Europe)** |
| 10 year bond | 0.76 | 0.55 | 0.56 | 3-month rates | 0.52 |
| 1 year swap | 0.58 | 0.37 | 0.39 | Term structure slopes | 0.60 |
| 5 year swap | 0.68 | 0.50 | 0.53 | VIX, VSTOXX | 0.87 |
| 10 year swap | 0.70 | 0.54 | 0.55 | ViTraxx, VCDX | 0.67 |

Note: The table shows linear correlations between some pairs of economic and financial indicators described in the Data section.

For each firm in a given sector, we estimate a regression explaining weekly variations in CDS spreads using either one of the synthetic factors described above: $CDS_{it} = \beta_0 + \beta_1 F_t, t = 1, 2, ..., T$, where $F_t$ denotes, alternatively, the Equity, Risk aversion, Macroeconomic, or Financial synthetic factor.[8] To make coefficient estimates comparable, we have standardized the synthetic indicators by subtracting their sample mean and dividing by their standard deviation. Thus, these regressions provide us with beta estimates for each firm on the four synthetic factors: equity, macroeconomic, risk aversion, and financial. In the four sectors considered, the coefficients on the equity factor and macroeconomic factors are negative, while those on the risk aversion factors are positive. Bull periods in stock markets reduce the probability of default and CDS spreads. Strong economic growth comes together with policy rates on the rise (so, a higher macro factor), and lower CDS spreads. An increase in the risk aversion factor (that is, on implied volatilites) means a higher perception of future uncertainty, and increased CDS spreads. Estimated coefficients on the financial factor are positive, but a number of firms in each sector have a negative coefficient. We calculate for each factor the correlation, across the firms in a given sector, between the absolute values of the estimated betas for that factor and our estimates of the relevance of each type of risk, with the results shown in Table 19.[9]

---

8    These are again least-squares, single equation estimates.
9    With a number of firms between 26 and 52 in each sector, statistical significance would require correlation coefficients above 0.30 in absolute value.

**Table 19.** Correlations between the size of the risk components and the sensitivity to the synthetic risk factors.

| Risk Component | MSCI | Risk Aversion | Macro | Financial |
|---|---|---|---|---|
| **European Industrial** | | | | |
| Systemic | 0.27 | 0.55 | 0.26 | −0.22 |
| Sectorial | 0.40 | 0.50 | 0.53 | −0.54 |
| Idiosyncratic | −0.36 | −0.63 | −0.41 | 0.38 |
| **US Industrial** | | | | |
| Systemic | 0.63 | 0.57 | 0.01 | −0.28 |
| Sectorial | 0.51 | 0.50 | 0.24 | −0.27 |
| Idiosyncratic | −0.64 | −0.60 | −0.11 | 0.31 |
| **European Financial** | | | | |
| Systemic | 0.64 | 0.63 | 0.39 | 0.12 |
| Sectorial | 0.21 | 0.71 | 0.90 | 0.16 |
| Idiosyncratic | −0.48 | −0.81 | −0.80 | −0.18 |
| **US Financial** | | | | |
| Systemic | 0.64 | 0.38 | 0.17 | 0.04 |
| Sectorial | 0.60 | 0.62 | 0.74 | 0.46 |
| Idiosyncratic | −0.78 | −0.59 | −0.49 | −0.26 |

Note: The table shows the linear correlation, across the firms in a given sector, between the absolute values of the coefficients estimated in the regression of weekly changes in CDS spreads on the synthetic factors, and the estimated size of each risk component: systemic, sectorial and idiosyncratic.

Systemic firms should be expected to react to events affecting the global situation of the economy, while firms with a large idiosyncratic component of risk should react to firm-specific news related to their behavior in the stock market, or news on their own accounting data, and not so much to global events. All the indicators we have been using in this paper have a global market nature, so that they should influence the more systemic firms, and not so much the more idiosyncratic ones. The same could be said of the synthetic factors, since they are linear combinations of the individual indicators.[10] In addition, we should bear in mind that Table 19 shows correlations with the estimated factor sensitivities, not with the synthetic risk factors themselves. Thus, the negative correlations across firms between the relevance of the idiosyncratic component of risk and the equity and the macroeconomic factors means that the CDS spread of the more idiosyncratic firms does not react much to stock market events or to changes in the macroeconomy. The latter may be due to the fact that the macroeconomic factor esentially captures the level of medium- and long-term interest rates, which by themselves do not signal any particular scenario for the more idiosyncratic firms, possibly subject to some specific financing conditions.

On the contrary, CDS spreads from systemic firms are very responsive to changes in the stock market or to the market forward perception of risk, as captured by implied volatility indicators, which are the main determinant of the risk aversion factor. CDS spreads from firms with a high sectorial component of risk are again sensitive to the stock market and also to forward-looking perceptions of uncertainty as captured by the risk aversion factor. In the US financial sector, the sectorial risk component is also related to the financial synthetic factor.

## 9. Conclusions

Whether or not the failure of a single firm evolves into a systemic crisis depends on the relevance of each firm in a given sector, as well as on the relevance of each sector in the global economy. In this paper, we have advanced a decomposition of credit risk at the level of sectors into a systemic and an idiosyncratic component. At the level of individual firms, we decompose credit risk among systemic,

---

10    Had we used firm-specific indicators, like the price of the firm's stock, or some indicator of the accounting results of the firm, we would expect to have the more idiosyncratic firms to be very responsive to such indicators.

sectorial and idiosyncratic components, an extremely useful decomposition for risk management because of the large-portfolio properties of idiosyncratic risk. As a portfolio becomes more granular, idiosyncratic risk is diversified away at the portfolio level. In the limit, when a portfolio becomes "infinitely fine-grained", idiosyncratic risk vanishes at the portfolio level, and only systematic and sectorial risk remains.

Our decomposition rests on the identification of a global credit risk factor, estimated as the first principal component of the 11 sectorial CDS indices, which we construct previously. We have shown that the information provided by the estimate is qualitatively invariante to alternative approaches to the estimation of such global risk factor. The information provided by this analysis has helped us to implement the risk decompositions mentioned above. We have identified the consumer goods and industrial sectors as being the most systemic. Health care and technology are the sectors displaying a higher idiosyncratic component of risk and, therefore, a lower correlation with all the other sectors. We have shown that well diversified credit portfolios with CDSs from a given sector have good possibilities for hedging by taking a contrary position in iTraxx or CDX indices or in their derivative products. The systemic and sectorial components explain around 65% of credit risk in the European industrial and financial firms, and 50% in the North American firms of those sectors, with 35% and 50% of credit risk, respectively, being idiosyncratic, which leaves a significant margin for portfolio diversification. The fact that idiosyncratic components of risk are larger in North American than in European firms may be due to a lack of liquidity.

Our analysis provides an element for a risk appetite framework at financial institutions, since they could easily use the numerical estimates of risk components we propose to maintain their risk limits when taking asset allocation decisions. Indeed, we have shown evidence suggesting that portfolios made up of firms with higher idiosyncratic components are easier to hedge, contrary to what happens with portfolios made up of firms with lower idiosyncratic risk components. This is observed uniformly over the industrial and financial sectors of Europe and North America. Furthermore, by evaluating the firms and sectors with the highest potential to produce systemic risk problems, our analysis should also be considered to be crucial for supervisors and regulators. Finally, we have explored the nature of each estimated risk component by analyzing its sensitivity to some synthetic risk factors. We have shown systemic firms to react to events affecting the global situation of the economy, having a higher sensitivity to risk factors based on stock market prices, the perception of future risk as captured by implied volatilities, or some business cycle indicators. We have also shown that the more idiosyncratic the firm, the less responsive it is to changes in these global factors. Both results support our decomposition of credit risk.

Additionally, our analysis has clear implications for credit risk management, since the sectorial strategy should depend on the risk decomposition of firms in a given sector. Indeed, it would seem appropriate to impose a maximum exposure to sectors where firms have a large systematic risk component while being relatively flexible about the distribution inside the sector, since a small idiosyncratic component would not allow us to extract the benefits of diversification by increasing the number of firms in the portfolio. On the contrary, in a sector where firms have a large idiosyncratic risk component, we should avoid having a high name concentration, since a better diversification would reduce the total risk of the portfolio.

We have restricted our analysis to firms on which CDS contracts have been issued. Further research should attempt to relate our estimated risk components to firms' characteristics such as size of assets and liabilities, profit and loss results, equity and bond prices and market share. That would allow for extending the evaluation of credit risk components for any other firm, even with no CDS contracs inssued on its name.

**Author Contributions:** A.C. provided the initial motivation for this research. Both authors contributed with ideas to the design of the empirical analysis, performed the statistical analysis, and wrote the successive versions of the paper.

**Funding:** Financial support by grants ECO2015-67305-P, PrometeoII/2013/015, Programa de Ayudas a la Investigación from Banco de España is gratefully acknowledged.

**Acknowledgments:** This article is an enlarged version of Chapter 3 in Chamizo's doctoral disssertation, available at https://eprints.ucm.es/40767/1/T38233.pdf. It reflects the opinions of the authors, but not the opinion of BBVA. The authors acknowledge comments received from L. Cimpean and J. De Juan Herreros.

**Conflicts of Interest:** The authors declare no conflict of interest.

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
