# Peer review of "Splitting Credit Risk into Systemic, Sectorial and Idiosyncratic Components"

_jrfm, doi:10.3390/jrfm12030129_

Round 1
Reviewer 1 Report
The general perspective on the manuscript hints to a fair academic effort and a well-written composition. The research question holds intrinsic value, and approach announced in the abstract and introduction is at first glance consistent enough to provide a relevant contribution. Despite these positive aspects, what makes an average composition become a good one is in general a rigorous approach and structure. My main observation with relation to this manuscript is that thought consistent, it needs a serious effort in restructuring and rewriting as some sections fail to serve the normal purposes and canons of academic writing.
I therefore invite the author/authors to consider the bellow observations:
1. The introduction is at best passable. I would have expected more into the direction of highlighting original, value-adding contributions.
2. The data section is oddly constructed. Though I’ve always advocated in the direction of providing a sufficient level of detail in academic writing, I personally do not consider that explaining the idea of ticker is anything but redundant. I advise towards restructuring the beginning section by removing such aspects and focusing on value-adding information. Visual elements such as descriptive statistics would represent a bonus (in addition to Table 1). Motivating the choice of items (for example risk aversion indicators) by previous studies in the literature would again represent a bonus.
3. Section 4 is a bit hectic. Again, I would hint towards a bit of restructuring with a focus on 4.2. This should be done with the purpose of putting forward a clear-cut methodological discussion. Introducing a separate section that deals solely with method might do the job.
4. Section 5 seems to be dedicated to results. It should be composed in this manner. I have some doubts about the effectiveness of figures 5.1 and 5.2, but this is only a minor comment.
5. The same applies to section 6.
6. Any potential robustness exercise would represent again a bonus.
All in all, the manuscript is an industrious effort, with an interesting approach that could become publishable given several transformations that do not require many changes in the general logic of the article.
Author Response
We want to sincerely thank the reviewers for their comments and suggestions. They made us think deeply about the contributions of the paper and about the best way to convey the information emerging from our analysis. As a result, the paper has been extensively restructured, and we believe that the revised version is a very significant improvement over the original version. We enclose a Word version of the paper where the extent of the changes introduced can be easily seen.
1. The introduction is at best passable. I would have expected more into the direction of highlighting original, value-adding contributions.
Response: The Introduction has been thoroughly rewritten, making an important effort to transmit the arguments underlying our analysis and emphasizing the contributions of our paper to the existing literature.
2. The data section is oddly constructed. Though I’ve always advocated in the direction of providing a sufficient level of detail in academic writing, I personally do not consider that explaining the idea of ticker is anything but redundant. I advise towards restructuring the beginning section by removing such aspects and focusing on value-adding information. Visual elements such as descriptive statistics would represent a bonus (in addition to Table 1). Motivating the choice of items (for example risk aversion indicators) by previous studies in the literature would again represent a bonus.
Response: The data section has been rewritten, eliminating the unneeded details and adding a motivation of our choice or risk aversion indicators. We have not done the same for other indicators because they are more standard in the literature. We have added tables showing the main statistics for all the indicators both in levels as well as in first differences. The data description is now embedded in a section where we construct and analyze the time evolution of sectorial credit indices.
3. Section 4 is a bit hectic. Again, I would hint towards a bit of restructuring with a focus on 4.2. This should be done with the purpose of putting forward a clear-cut methodological discussion. Introducing a separate section that deals solely with method might do the job.
Response: Section 4, 5 and 6 of the first version of the paper have been completely restructured.
Section 4 is now devoted to present the results from the estimation of the Global Risk Factor, and to analyze its information content on the credit market. We compare that information to the one provided by other risk indicators. As suggested by the reviewer, Section 4 begins with a description of the methodology used for the estimation of the Global Risk Factor. Being much shorter, the content of this section should now be much better focused.
4. Section 5 seems to be dedicated to results. It should be composed in this manner. I have some doubts about the effectiveness of figures 5.1 and 5.2, but this is only a minor comment.
Response: Section 5 is fully devoted to the single issue of risk decomposition at the level of sectors and at the level of the individual firms. Thus, its role in the line of reasoning of the paper can be much better grasped now. We have eliminated Figure 5.2.
5. The same applies to section 6.
Response: This section is now devoted to presenting validation tests for our risk decomposition methodology, in the form of i) the properties of the implied idiosyncratic risk components, ii) the hedging properties of portfolios with different degree of idiosyncratic risk, and iii) the relationship between idiosyncratic risk and lack of liquidity. The point of the section is to analyze whether the estimated idiosyncratic risk components have the properties to be expected from such components
6. Any potential robustness exercise would represent again a bonus.
Response: We have not performed a robustness exercise in the sense of considering alternative constructions of our Global Risk Factor. Some possibilities, like considering more than one principal component do not improve upon the already good information content of our Global Risk Factor estimate and they would enlarge an already long paper with not much gain to the reader. On the other hand, we believe that the different exercises in Section 6, as described above, are a good substitute of robustness exercises, since we examine in detail whether the risk components that come out of our risk decomposition methodology fulfill the conditions that should be expected from their nature.
Reviewer 2 Report
jrfm-548124
The paper provides a very useful method to decompose the systemic, sectoral and idiosyncratic components of credit risk. Also, the methodology is simple to compute and can be applied to different financial markets. The authors use a principal component analysis and regression models to try to get deep into the empirical implications of their findings. Overall this is a nice draft.
However, there are a few points I would like the authors to consider (and add) in addition to the analysis they have already carried out.
1. An important paper must be including, Ballester et al (2016) that propose an innovative framework to distinguish between two types of contagion: systematic and idiosyncratic for the financial sector. Also, the authors apply principal component analysis (PCA) to extract the common patterns underlying the correlations among the CDS returns of individual banks.
This is the same procedure that in this paper is carry-out, therefore is important to cite.
2. Figures: The y-axis of Figure 4.1. maybe is wrong; the x-axis of Figure 5.2 is not well clear, also why the y-axis shows 120%? 1-r^2 is always less than 100%
3. The methodologies (derivation, formulas, etc) are complete absence. I suggest implementing it, in order to better explain the type of factor analysis or PCA (for example the choice of a number of components), as well as the type of regression (for example the results in Table 3). What kind of regression is applied? Panel model? OLS estimation?
4. I suggest adding several tables to better explain the results. For example, the results of PCA (see page 11, “Cumulative percentage of total…”), of the residuals of regressions (please see page 15, “Further evidence…”; please see page 17, “We could of the residuals…”). Also, in section 6.4. to highlight the correlation analysis (please see page 26, “Some indicators display…”)
5. To strengthen the analysis, I suggest adding residual diagnostic tests (for example, the normality test)
6. Section 5: which windows time is performed the rolling window correlation? I don’t understand.
I wish the authors all the best for future improvements.
References
Ballester, L., Casu, B., & González-Urteaga, A. (2016). Bank fragility and contagion: Evidence from the bank CDS market. Journal of Empirical Finance, 38, 394-416.

Author Response
Reviewer 2
The paper provides a very useful method to decompose the systemic, sectoral and idiosyncratic components of credit risk. Also, the methodology is simple to compute and can be applied to different financial markets. The authors use a principal component analysis and regression models to try to get deep into the empirical implications of their findings. Overall this is a nice draft.
However, there are a few points I would like the authors to consider (and add) in addition to the analysis they have already carried out.
General response: We want to sincerely thank the reviewers for their comments and suggestions. They made us think deeply about the contributions of the paper and about the best way to convey the information emerging from our analysis. As a result, the paper has been extensively restructured, and we believe that the revised version is a very significant improvement over the original version. We enclose a Word version of the paper where the extent of the changes introduced can be easily seen.
1. An important paper must be including, Ballester et al (2016) that propose an innovative framework to distinguish between two types of contagion: systematic and idiosyncratic for the financial sector. Also, the authors apply principal component analysis (PCA) to extract the common patterns underlying the correlations among the CDS returns of individual banks. This is the same procedure that in this paper is carry-out, therefore is important to cite.
Response: References are now made to the Ballester et al. (2016) paper.
2. Figures: The y-axis of Figure 4.1. maybe is wrong; the x-axis of Figure 5.2 is not well clear, also why the y-axis shows 120%? 1-r^2 is always less than 100%
Response: The figures have been fixed to clarify legends, axis titles and axis values.
3. The methodologies (derivation, formulas, etc) are complete absence. I suggest implementing it, in order to better explain the type of factor analysis or PCA (for example the choice of a number of components), as well as the type of regression (for example the results in Table 3). What kind of regression is applied? Panel model? OLS estimation?
Response: Section 4 contains a description of the PCA methodology used in our analysis. We have also added the analytical expression of the different regression models used throughout the paper. For each estimated model, we have added a statement describing the fact that these are least-squares, single equation estimates. In most cases (although not in all of them), the explanatory variables are the same across the different models and thus, there would not be any change if we followed a simultaneous equation estimation strategy.
4. I suggest adding several tables to better explain the results. For example, the results of PCA (see page 11, “Cumulative percentage of total…”), of the residuals of regressions (please see page 15, “Further evidence…”; please see page 17, “We could of the residuals…”). Also, in section 6.4. to highlight the correlation analysis (please see page 26, “Some indicators display…”)
Response: We have added tables describing some of the results obtained in our analysis. Tables 1 and 2 contain the main statistics for the wide set of indicators used in the paper, and for their first difference. Table 6 shows the main results obtained from the application of the principal component methodology to the set of time series of sectorial credit indices. Table 7 contains the relevant correlations between indicators and the component of the Global Risk factor that is orthogonal to iTraxx.
5. To strengthen the analysis, I suggest adding residual diagnostic tests (for example, the normality test).
Response: We have carefully considered the possibility of adding residual diagnostic tests for our estimated regressions, but we have decided against doing that because it would increase significantly the size and number of tables in the paper, especially taking into account that we have added 4 new tables to the revised version. An examination of the structure and information content of our tables quickly indicates that adding diagnostic tests for residual autocorrelation, heteroscedasticity, and normality would require a notorious increase in size or a split of each table of regression estimates into at least two tables. Such checks are clearly interesting, but they point out to a potential lack of efficiency, more than to any bias or any spurious relationship. We have chosen to include information on possible nonstationarity of residuals because that could be easily done and lack of stationarity would be very damaging to the use of our estimate models for risk management.
6. Section 5: which windows time is performed the rolling window correlation? I don’t understand.
Response: It was estimated using annual rolling windows (52-week data points). This is now explained in the paper.
Round 2
Reviewer 1 Report
Firstly, I congratulate the author/authors for their efforts in the direction of providing a better manuscript.
Comment 1: The changes made to the introduction are passable and the section is in far better shape than the original version.
Comment 2: The upgrades in the data section are suitable and add to the cohesion of the manuscript.
Comment 3,4 and 5: The author/authors is/are successful in upgrading the sections in discussion to a certain extent. Despite the fact that I expected a more substantial restructuring, I consider the present version as passable.
Comment 6: The hint towards robustness testing was central to my initial evaluation of the manuscript in the sense that it represented the main value-adding recommendation. I am deeply unconvinced by the fact that the different exercises in section 6 could be regarded as a sufficient substitute for robustness testing. However, given the initial positive impression on the article and efforts put in providing a more refined version I will not insist on this any further.
Author Response
General response: We thank the Reviewer for the effort in going through the previous version of our paper and making suggestions to improve the presentation. We have done our best, in a short period of time, to satisfy them.
Reviewer 1
Firstly, I congratulate the author/authors for their efforts in the direction of providing a better manuscript.
Comment 1: The changes made to the introduction are passable and the section is in far better shape than the original version.
Response: We have just modified a few details in this version
Comment 2: The upgrades in the data section are suitable and add to the cohesion of the manuscript.
Response: We have introduced a more complete Data description in this version
Comment 3,4 and 5: The author/authors is/are successful in upgrading the sections in discussion to a certain extent. Despite the fact that I expected a more substantial restructuring, I consider the present version as passable.
Response: Following the recommendation from reviewer 2, we have introduced in these sections some formulae representing the models being estimated in them. We have also introduced much more information on the estimation output of different models.
Comment 6: The hint towards robustness testing was central to my initial evaluation of the manuscript in the sense that it represented the main value-adding recommendation. I am deeply unconvinced by the fact that the different exercises in section 6 could be regarded as a sufficient substitute for robustness testing. However, given the initial positive impression on the article and efforts put in providing a more refined version I will not insist on this any further.
Response: We have introduced a new Section 7 on Robustness, analyzing the changes that would be observed if our estimation proposal of a Global Risk Factor is modified: i) by estimating principal components directly from CDS spread data in levels, ii) by excluding the financial and government sectors. None of these changes introduces any qualitative change in our results, and quantitative results are very similar.
Notes:
1.- We have taken the opportunity to modify our synthetic factor analysis in Section 8 reducing the number of factors from 4 to 3. All the results in this section have been obtained under this new formulation. We believe that it clarifies the discussion and the presentation of results.
Reviewer 2 Report
Please see the attached file
all the best

Author Response
General response: We thank the Reviewer for the effort in going through the previous version of our paper and making suggestions to improve the presentation. We have done our best, in a short period of time, to satisfy them.
Reviewer 2
Now the paper seems much better structured, i.e. the authors have modified the paper as suggested. However, there are still shortcomings.
1. I suggested (see point 5): To strengthen the analysis, I suggest adding residual diagnostic tests (for example, the normality test).
Response of authors: We have carefully considered the possibility of adding residual diagnostic tests for our estimated regressions, but we have decided against doing that because it would increase significantly the size and number of tables in the paper, especially taking into account that we have added 4 new tables to the revised version. An examination of the structure and information content of our tables quickly indicates that adding diagnostic tests for residual autocorrelation, heteroscedasticity, and normality would require a notorious increase in size or a split of each table of regression estimates into at least two tables. Such checks are clearly interesting, but they point out to a potential lack of efficiency, more than to any bias or any spurious relationship. We have chosen to include information on possible nonstationarity of residuals because that could be easily done and lack of stationarity would be very damaging to the use of our estimate models for risk management.
However, I don’t agree with the response of the authors. I don't think adding tables is a problem, on the contrary. The analysis helps to better clarify and strengthen the results of the analysis. I suggest adding a few more explanations of the residuals.
Response: We have introduced normality tests, tests for autocorrelation at various lags, ARCH tests and unit root tests in the residuals from our estimated regression models.
2. Methodology is better explained, but there are still gaps.
a. Section: 3.2.3. The equation is not explained. There is a footnote that explains the estimation method, but the equation is still missing.
Response: Here, and at several other points throughout the paper we have introduced explicit formulae for the models we estimate. We have also been more explicit about the estimation method, which is always single equation, least-squares.
b. I recommend entering the equation number (in succession). For example (see section 4.1.)
PC=XW (n.)
Response: Here, and throughout the paper we have introduced equation numbers where needed
c. And to other equations (see section 4.2., GRF formula; section 5.2, CDS formulas;
Response: We have an effort to explain better the models to be estimated, introduce explicit analytical formulae, and provide more statistical information regarding the estimation output
Notes:
1.- We have taken the opportunity to modify our synthetic factor analysis in Section 8 reducing the number of factors from 4 to 3. All the results in this section have been obtained under this new formulation. We believe that it clarifies the discussion and the presentation of results.
2.- We have introduced a more complete Data description in this version
3.- Following a suggestion from a comment made by Reviewer 1, we have added a new Section 7 on Robustness